# The Wg and Dpp morphogens regulate gene expression by modulating the frequency of transcriptional bursts

Rachael Bakker[1,2], Madhav Mani[1,2,3]*, Richard W Carthew[1,2]*

[1]Department of Molecular Biosciences, Northwestern University, Evanston, United States; [2]NSF-Simons Center for Quantitative Biology, Northwestern University, Evanston, United States; [3]Department of Engineering Sciences and Applied Mathematics, Northwestern University, Evanston, United States

**Abstract** Morphogen signaling contributes to the patterned spatiotemporal expression of genes during development. One mode of regulation of signaling-responsive genes is at the level of transcription. Single-cell quantitative studies of transcription have revealed that transcription occurs intermittently, in bursts. Although the effects of many gene regulatory mechanisms on transcriptional bursting have been studied, it remains unclear how morphogen gradients affect this dynamic property of downstream genes. Here we have adapted single molecule fluorescence in situ hybridization (smFISH) for use in the *Drosophila* wing imaginal disc in order to measure nascent and mature mRNA of genes downstream of the Wg and Dpp morphogen gradients. We compared our experimental results with predictions from stochastic models of transcription, which indicated that the transcription levels of these genes appear to share a common method of control via burst frequency modulation. Our data help further elucidate the link between developmental gene regulatory mechanisms and transcriptional bursting.

*For correspondence:
madhav.mani@northwestern.edu
(MM);
r-carthew@northwestern.edu
(RWC)

Competing interests: The authors declare that no competing interests exist.

## Introduction

Paracrine signaling is a highly conserved means for cells within a tissue to communicate with one another to regulate diverse activities including proliferation, differentiation, apoptosis, and movement. Many of these activities are mediated by changes in gene transcription that are brought about by reception of the signals. Paracrine factors acting as morphogens are a particularly important class of gene regulators. Morphogens form spatially-extended gradients from the source of their synthesis, and elicit different transcription outputs from target genes, depending on local concentration of the morphogen (*Tabata and Takei, 2004*). Many paracrine signals regulate gene transcription via control of the availability or activity of sequence-specific transcription factors. Some transcription factors regulate assembly of the preinitiation complex composed of Pol II and general factors at the transcription start site (*Esnault et al., 2008*). Other factors recruit coregulators that modify nucleosomes or remodel the chromatin architecture of the gene (*Bannister and Kouzarides, 2011*).

However, transcription is a dynamic process, and thus, molecular models of regulation via PIC assembly or chromatin structure, do not adequately capture what kinetic steps in transcription initiation are being regulated. Recently developed methods have uncovered greater complexity in the transcription initiation process than previously imagined. Genes that are constitutively expressed rarely show uniform and continuous mRNA synthesis. Rather, mRNA synthesis occurs in bursts that are interrupted by periods of dormant output. This phenomenon is known as transcriptional bursting (*Chen et al., 2019*; *Chubb et al., 2006*; *Dey et al., 2015*; *Raj et al., 2006*; *Suter et al., 2011*).

Various studies have explored how mechanisms of gene regulation affect the size and frequency of transcriptional bursts, and thereby affect transcription output. The availability of transcription factors has been shown to affect burst frequency (*Ezer et al., 2016*; *Larson et al., 2013*; *Senecal et al., 2014*). For example, the *Drosophila* transcription factors Bicoid and Dorsal have been studied in great detail with respect to their effects on transcription burst frequency in the embryo (*Garcia et al., 2013*; *He et al., 2012*; *Holloway and Spirov, 2017*; *Little et al., 2013*; *Xu et al., 2015*). Enhancer strength and enhancer-promoter contact correlate with burst frequency of genes (*Bartman et al., 2016*; *Bothma et al., 2014*; *Chen et al., 2019*; *Fukaya et al., 2016*; *Larsson et al., 2019*). These studies altogether suggest that bursting frequency is potentiated by enhancer-promoter contact and is mediated by transcription factors binding to DNA.

In this study, we have explored how the Wnt protein Wingless (Wg) and BMP protein Decapentaplegic (Dpp) regulate transcription dynamics of genes in the *Drosophila* wing imaginal disc. The Wnt and BMP families of proteins are two highly conserved paracrine factors that can act as morphogens. In canonical Wnt signaling, the binding of extracellular Wnt protein to its transmembrane receptor Frizzled causes β-catenin to be stabilized and free to enter the nucleus, where it relieves repression of Wnt-responsive genes by binding to the sequence-specific transcription factor TCF (*Clevers and Nusse, 2012*; *Swarup and Verheyen, 2012*). In canonical BMP signaling, ligand binding to receptor triggers phosphorylation of SMAD proteins, which translocate to the nucleus along with co-SMADs, bind to responsive genes, and activate their transcription (*Hamaratoglu et al., 2014*; *Shi and Massagué, 2003*).

To explore the effects of Dpp and Wg signaling on transcription dynamics, we have adapted single molecule fluorescent in situ hybridization (smFISH) for use in imaginal disc tissues. We use smFISH to quantify nascent and mature mRNAs for several genes expressed in highly diverse spatial patterns within the wing disc. Taken together, our data suggest that all of the genes investigated are regulated by modulation of their transcription burst frequency by Dpp and Wg even though their mean expression patterns are distinct from one another.

## Results

In this study, we have explored how the Wg and Dpp morphogens regulate transcription dynamics in the wing disc. Each morphogen is synthesized in a narrow stripe of cells within the disc. Wg is produced in cells at the boundary between Dorsal and Ventral (DV) compartments of the wing pouch, while Dpp is produced in cells at the boundary between Anterior and Posterior (AP) compartments (*Figure 1A*). These factors form concentration gradients across the disc, and in the case of Dpp, it regulates gene expression in a concentration-dependent manner.

### smFISH detection of mRNA molecules in the wing disc

In order to assay gene expression in the wing imaginal disc, we quantified mRNA numbers using smFISH. With smFISH, a tandem array of fluorescently-labeled oligonucleotides complementary to a given mRNA are hybridized to fixed and permeabilized tissue. When a sufficient number of oligo probes anneal to one mRNA molecule, the aggregate fluorescence can be detected by standard confocal microscopy (*Raj et al., 2008*). This method has been developed and applied to many systems, including cell culture, *C. elegans*, and the *Drosophila* embryo (*Ji and van Oudenaarden, 2012*; *Little and Gregor, 2018*; *Youk et al., 2010*). We developed a robust smFISH method applicable for imaginal discs (see Materials and methods for details).

We first probed for expression of the *senseless* (*sens*) gene in the wing disc. *Sens* is required for cells to adopt a sensory organ fate, and the gene is expressed in two stripes of cells adjacent to and on either side of the DV boundary in the wing pouch (*Figure 1B,C*; *Nolo et al., 2000*). *Sens* expression in the wing pouch is induced by Wg, which is expressed by cells located at the DV boundary (*Jafar-Nejad et al., 2006*). We probed for *sens* mRNAs expressed from a transgenic version of the *sens* gene. We did so for a number of reasons. First, the genomic transgene rescues the endogenous gene based on function and expression (*Cassidy et al., 2013*). Second, the transgene is tagged such that the amino-terminal coding sequence corresponds to super-fold GFP (sfGFP). By using oligo probes directed against sfGFP, we could easily determine the specificity of detection.

Discs from *sfGFP-sens* animals were probed and imaged by confocal microscopy, revealing the expected pattern of fluorescence localized to two stripes adjacent to the DV midline in the wing

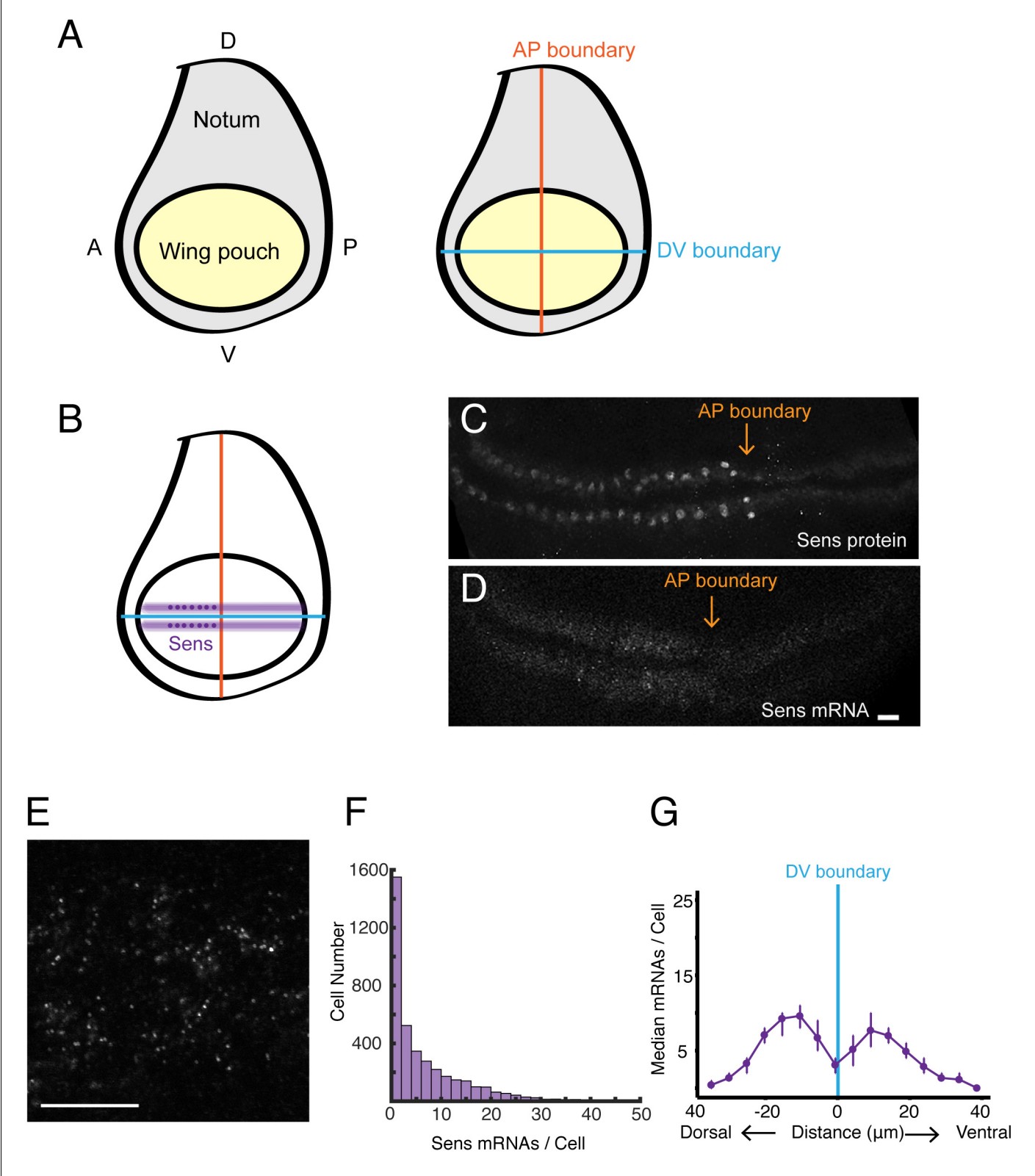

**Figure 1.** smFISH analysis of sfGFP-sens mRNA levels in wing imaginal discs. (**A**) Schematic of a wing disc outlining different regional domains, and the positions of boundaries between Dorsal (D) - Ventral (V) and Anterior (A) - Posterior (P) compartments of the disc. Each wing disc is composed of roughly 50,000 cells organized in a pseudostratified epithelium. (**B**) Schematized expression pattern for Sens inside the wing pouch centered around the
*Figure 1 continued on next page*

*Figure 1 continued*

DV boundary. Sens is also expressed in clusters of cells in the notum, which are not shown. (**C-E**) Confocal sections of wing discs expressing sfGFP-Sens. (**C**) sfGFP-Sens protein fluorescence. (**D**) sfGFP-Sens mRNAs as visualized by smFISH using sfGFP probes. Scale bar = 10 µm. (**E**) Higher magnification of sfGFP-Sens mRNAs as visualized by smFISH using sfGFP probes. Scale bar = 10 µm. (**F**) Distribution of wing disc cells as a function of the number of Sens mRNA molecules per cell. (**G**) Sens mRNA number as a function of cell distance from the DV boundary displays a bimodal expression pattern for Sens. Cells were binned according to the shortest path length from its centroid to the DV boundary, and whether they were dorsal or ventral compartment cells. Median mRNA number/cell for each bin is plotted with 95% bootstrapped confidence intervals.

The online version of this article includes the following figure supplement(s) for figure 1:

**Figure supplement 1.** Development of smFISH imaging and analysis.
**Figure supplement 2.** Determination of false-positive and false-negative rates for smFISH.
**Figure supplement 3.** smFISH imaging of the eye imaginal disc.

pouch (*Figure 1D*). The fluorescence signal was specific for *sfGFP-sens* since wing discs from larvae not carrying the transgene gave a low background fluorescence pattern (*Figure 1—figure supplement 1A,B*). The fluorescence signal from *sfGFP-sens* discs was sufficiently bright that spots were readily detected in optical sections when imaged under higher magnification (*Figure 1E*). The size of each 2D spot was approximately the expected diffraction limit of ~600 nm for smFISH probes emitting at 633 nm wavelength (*Lipson et al., 1995*). A custom image-analysis pipeline was developed to segment and analyze all of the 3D fluorescent spots in an entire stack of optical sections (*Figure 1—figure supplement 1C*). Details of the segmentation and analysis are provided in the Materials and methods.

We tested the ability of the pipeline to correctly identify RNA spots by several means. First we expected *sfGFP-sens* mRNA molecules to generate fluorescence spots with a homogeneous composition since the mRNAs could equivalently anneal to the probes. The distribution of fluorescence intensity for the identified 3D spots was unimodal, suggesting that the spots had a homogeneous composition (*Figure 1—figure supplement 1D*). Second, we incubated wing discs in medium containing actinomycin-D, an inhibitor of mRNA synthesis. The number of fluorescence spots was greatly diminished, as would be expected if they were localized to mRNA molecules (*Figure 1—figure supplement 1E*). Third, if the method is accurate, almost all spots would correspond to *sfGFP-sens* mRNAs. We compared the number of identified spots in discs expressing the *sfGFP-sens* transgene versus discs lacking the transgene. From this, we estimated that 0.5% of identified spots are false-positive (*Figure 1—figure supplement 2A*). Finally, we estimated the number of sfGFP-sens mRNAs that fail to be identified as fluorescent spots. We simultaneously hybridized *sfGFP-sens* wing discs with two sets of non-overlapping probes - one set recognized *sfGFP* and the other set recognized *sens* sequences. Each probe-set was labeled with a different fluor. If a spot identified using the *sfGFP* probe set was not identified by the *sens* probe-set, we classified that spot as a false-negative. The analysis indicated that a maximum of 6% of mRNAs (232 out of 3842 spots scored) were not identified by both probe-sets (*Figure 1—figure supplement 2B*). This rate of false-negative identification is comparable to smFISH methods in other systems (*Raj et al., 2008*).

We next looked to partition identified mRNAs into the cells from which they were expressed. Since the smFISH method denatured the epitopes of all tested antibodies and it also denatured sfGFP, we were unable to segment cells using standard approaches. In the absence of a direct approach, we adopted a computational approach to resolving the smFISH signal at single-cell resolution. Using the fluorescent dye DAPI to visualize cell nuclei in the imaged samples, we segmented nuclei into 3D objects (*Figure 1—figure supplement 2C–E*), which are located throughout the apical-basal axis of the pseudostratified epithelium of the wing disc (*Aldaz and Escudero, 2010*). Based on segmented nuclei, we were able to construct effective cell boundaries by performing a 3D Voronoi tessellation (*Figure 1—figure supplement 2F*). RNAs were then partitioned into the distinct Voronoi cells (*Figure 1—figure supplement 2G*). Despite the local inaccuracies in our protocol for assigning transcripts to single cells, the Voronoi based tessellation of the three-dimensional tissue is a democratic prescription, lacking any hyperparameters, that is able to reveal the global quantitative trends in the data. The same democratic approach has been used by others in assigning mRNA transcripts to early embryonic nuclei when cell boundaries are unseen (*Little et al., 2013*). Details of tessellation are provided in the Materials and methods.

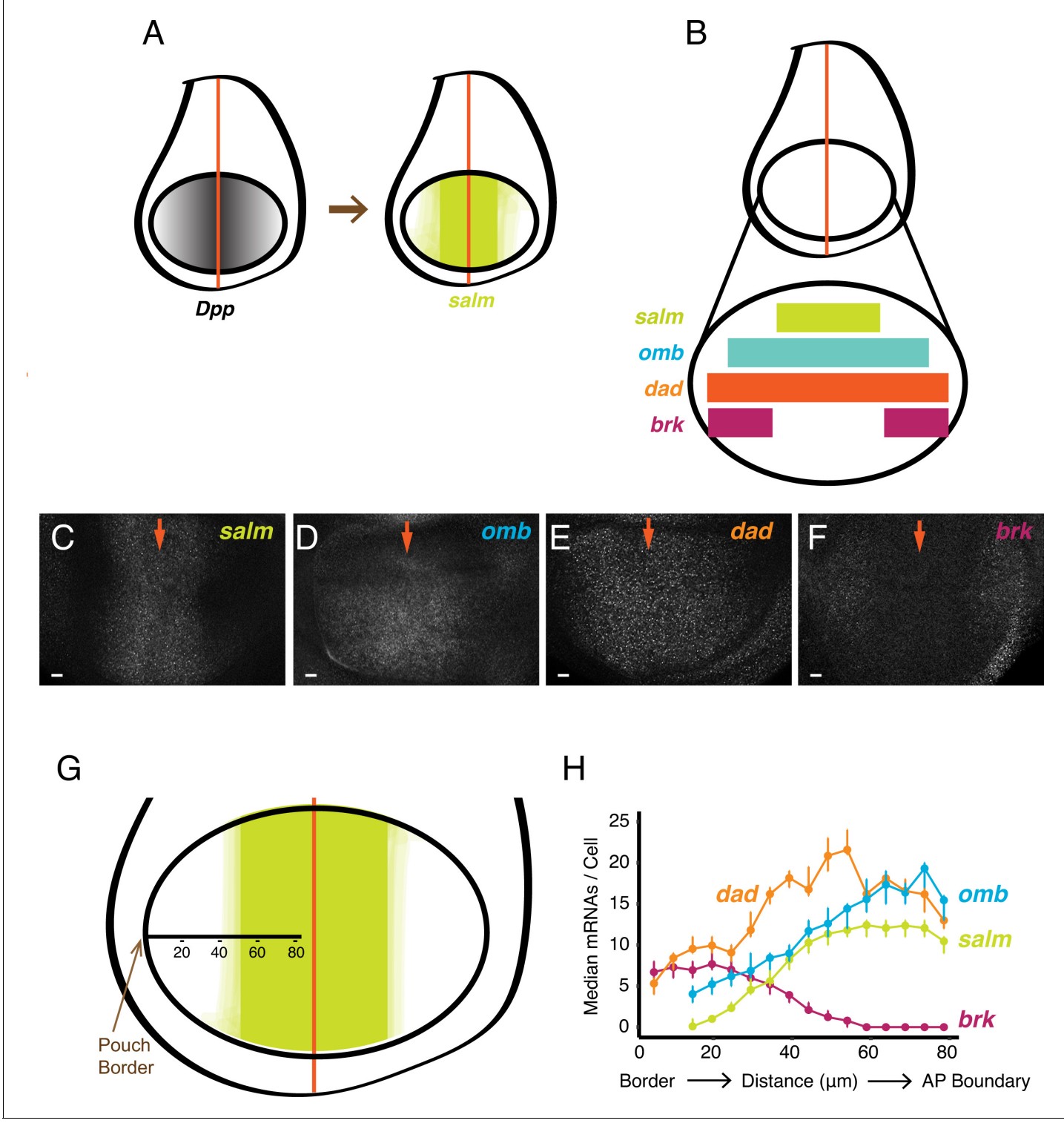

**Figure 2.** smFISH analysis of mRNA levels from Dpp-responsive genes. (A) Schematic of wing discs highlighting the graded distribution of Dpp protein in the wing pouch, centered around the AP boundary, and the expression domain for *salm*, one of the targets of Dpp regulation. Not shown is Dpp localization in the notum domain of the disc. (B) Expression domains of four target genes of Dpp signaling. (C-F) Confocal sections of wing pouches probed for mRNAs synthesized from the *salm* (C), *omb* (D), *dad* (E), and *brk* (F) genes. Orange arrows mark the position of the AP boundary in each image. (G, H) mRNA number as a function of cell distance from the anterior-most border of the wing pouch. (G) A border-to-boundary axis, orthogonal to the AP boundary, is used to map cell position, along which distances are displayed in μm from the wing pouch border. (H) Cells were binned

*Figure 2 continued on next page*

*Figure 2 continued*

according to position along the border-to-boundary axis. Median mRNA number/cell for each bin is plotted with 95% bootstrapped confidence intervals.

The online version of this article includes the following figure supplement(s) for figure 2:

**Figure supplement 1.** Detection of RNAs corresponding to the *sd* gene.

The abundance of *sens* mRNAs within the DV stripes varied from one to fifty molecules per cell (*Figure 1F*), reflecting the graded expression pattern of Sens protein induced by the Wg morphogen across the width of each stripe (*Jafar-Nejad et al., 2006*). Binning cells according to their distance from the DV boundary, we were able to observe peaks in mRNA number per cell as a function of distance from the boundary (*Figure 1G*).

We also used the *sfGFP-sens* gene to determine whether the smFISH method could detect mRNAs in other imaginal discs. In the eye disc, *sens* is expressed in a stripe of cells located within the morphogenetic furrow, and indeed we were able to detect smFISH signals in furrow cells of the eye disc (*Figure 1—figure supplement 3*). Thus, our method is broadly applicable to imaginal discs.

## smFISH detection of gene expression regulated by Dpp

We extended the analysis to genes downstream of the BMP family protein Dpp. Dpp is expressed in a stripe of cells located at the AP boundary of the wing disc, orthogonal to the Wg stripe (*Figure 2A*). Dpp protein is transported bidirectionally to form gradients across the disc, and several genes are regulated by Dpp in a concentration-dependent manner. S*palt-major* (*salm*), *optomoter-blind* (*omb*), *daughters-against-dpp* (*dad),* and *brinker* (*brk*) are expressed in symmetric domains within the anterior and posterior compartments of the wing pouch (*Figure 2A,B*). *Salm* is symmetrically expressed in a domain somewhat broader than the Dpp stripe, whereas *omb* and *dad* are expressed more broadly, and *brk* is expressed only near the wing pouch border (*de Celis et al., 1996*; *Grimm and Pflugfelder, 1996*; *Tabata and Takei, 2004*). When smFISH was used to detect mRNAs of these genes, it qualitatively recapitulated their known expression patterns (*Figure 2C–F*). We quantified the number of mRNAs per cell and attempted to map the distribution to cell position within the wing pouch. Since the only landmark we could reliably use was the border between the wing pouch and the rest of the disc, we measured cell position as a function of distance from the border (*Figure 2G*). When we did so, the distributions in mRNA number per cell displayed profiles that were consistent with previous qualitative descriptions of their expression patterns (*Figure 2H*). To ensure that these distributions were not an artifact of landmarking the border, we probed for mRNAs produced from the *scalloped* (*sd*) gene. The *sd* gene is expressed uniformly throughout the wing pouch (*Campbell et al., 1992*; *Williams et al., 1993*), and thus we anticipated a uniform distribution of mRNAs/cell if our method was accurate. Indeed, there was a fairly constant level of mRNAs/cell across the wing pouch as determined by our smFISH pipeline (*Figure 2—figure supplement 1A*).

## smFISH detects sites of nascent transcription

A further benefit to smFISH is that it can detect and quantify RNA as it is being transcribed from a gene. We sought to identify and characterize these sites of nascent transcription in the wing disc. Quantification of pixel intensity of all fluorescent spots revealed two discrete populations: a large population of dim spots of uniform intensity, and a smaller population of brighter spots with more variable intensity (*Figure 3A,B*). The former population corresponded to those described earlier, and they were primarily located in the cytoplasm - these are the mature mRNAs. The latter population was primarily located inside nuclei, and thus we hypothesized that these were sites of nascent transcription. To confirm that these bright spots corresponded to transcription sites, we used probes complementary to an intron in the *omb* gene. These probes only detected the brighter population of spots localized to nuclei (*Figure 3C*). Since introns are not spliced out until after transcription, this result supports the conclusion that the brighter nuclear spots are sites of nascent transcription.

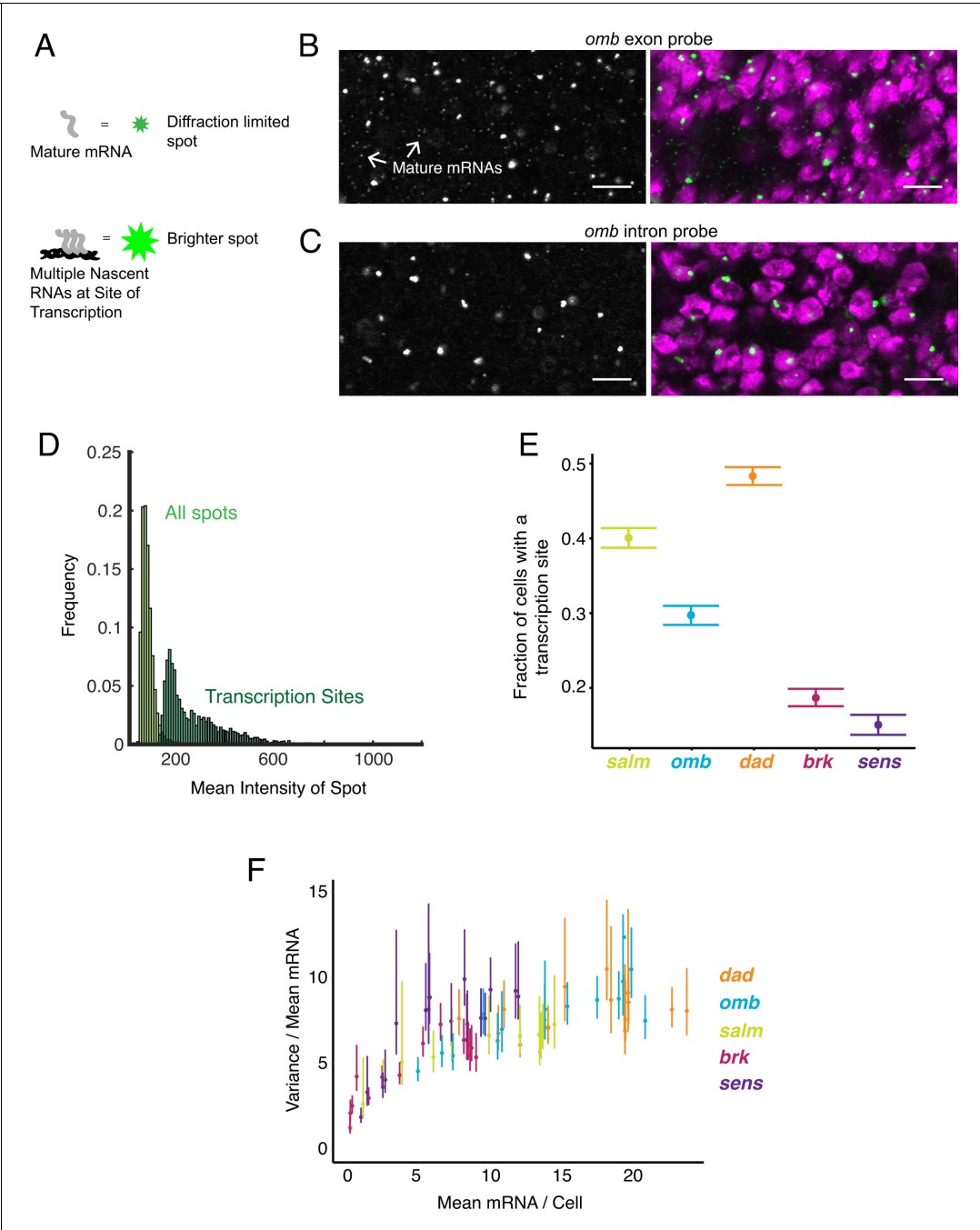

**Figure 3.** Sites of nascent transcription are detected by smFISH. (**A**) Sites of nascent transcription can fluoresce more brightly than single mRNA molecules due to multiple nascent transcripts localized to one gene locus. (**B**) Probes recognizing an *omb* exon generate many small dim spots and a few large bright spots. Right image shows the merge of probe and DAPI fluorescence. The bright spots are associated with nuclei whereas most dim spots are not. (**C**) Probes recognizing an *omb* intron only generate large bright spots that are associated with nuclei. Scale bars = 5 µm. (**D**) Frequency distribution of intensity for all spots identified in a wing disc probed for *sens* RNAs. Using a threshold of twice the median spot intensity, all single mRNA spots were filtered out, leaving only spots that are associated with transcription sites. The frequency distribution for this class of spots is shown. (**E**) Transcription sites are assigned to cells. For each cell that contains one or more mRNA molecules, it is scored for whether it also has one or more transcription sites. The average fraction of all such cells with a transcription site is shown for each gene. Error bars represent 95% confidence intervals. (**F**) The ratio of the variance of mRNAs/cell to its mean, as a function of the mean, for all genes. This ratio is larger than one, irrespective of the mRNA number for binned sub-populations of cells and the gene. Error bars represent 95% confidence intervals.

The online version of this article includes the following figure supplement(s) for figure 3:

**Figure supplement 1.** Detection of transcription sites and their quantification.

*Figure 3 continued on next page*

*Figure 3 continued*

**Figure supplement 2.** Transcription sites and mRNA patterns in unsegmented images.

Although wing disc cells are diploid, fewer than 15% of nuclei contained more than one transcription site for a given gene. One explanation is that transcription is infrequent enough that 85% of the time only one allele is actively transcribing. Another explanation is that two alleles are physically co-localized, and their nascent transcripts cannot be resolved by confocal microscopy. *Drosophila* and other animals have extensive physical pairing of homologous chromosomes in somatic cells (*McKee, 2004*). Consequently, alleles on paired chromosomes are often spatially juxtaposed (*Szabo et al., 2018*). For genes such as *omb* that we probed far upstream of the transcription termination site, it is likely that we were observing transcription from both alleles at once, given that a detectable nascent RNA would stay at the transcription site for a long time (~50 min). Even for these very bright transcription spots, only one transcription site per nucleus was observed (*Figure 3B,C*). This observation is consistent with a single transcription spot in a nucleus representing transcription from both alleles.

## Transcription occurs in bursts

Transcription sites were counted by applying a cutoff that only included spots with at least twice the intensity of a mature mRNA spot (*Figure 3D*, *Figure 3—figure supplement 1*). There was a broad distribution of transcription site intensities, suggesting a large range of nascent RNA numbers that were present on a gene at a given time.

Strikingly, many cells did not have a detectable transcription site even though the cells contained mature mRNAs (*Figure 3E*). Between 50–80% of all cells had this feature, and it was observed for all genes. This observation is not an artifact of segmentation erroneously assigning mature mRNAs to cells that do not express the genes. For all genes, the number of transcription sites strongly correlated with mRNA number when discs were binned but not segmented (*Figure 3—figure supplement 2*). Hence, although assignment errors occur at the local scale, they cannot account for the quantitative global trends where 2–5 fold more cells lack a transcription site than lack any mature mRNAs.

Why do cells with mature mRNAs lack detectable transcription sites? One explanation is that each gene's promoter is always open, but since transcription is stochastic, there would be times when zero or just a few Pol II molecules are presently transcribing the gene. In this scenario, the birth and death of mRNAs can be described as a Poisson process, where the ratio of the variance of the distribution of number of mRNAs to its mean is expected to be one (*Munsky et al., 2012*; *Raj and van Oudenaarden, 2008*). Since mRNA number per cell varied systematically across the wing disc because of Wg and Dpp signaling, we binned cells according to their position in the disc, as had been described earlier (*Figures 1G* and *2H*), and empirically estimated the ratio of a bin's variance to its mean. The ratio of variance to mean mature mRNA number per cell was between 5 and 10 for all genes, and was fairly independent of mRNA output (*Figure 3F*). This indicated that a Poisson process could not explain why we failed to detect transcription sites in every cell expressing mRNA.

To determine if our observations were possibly caused by transcription bursting, we invoked a two-state model of transcription (*Figure 4A*). A promoter exists in one of two possible states - ON and OFF. The promoter switches between states at particular rates $k_{on}$ and $k_{off}$. When the promoter is in the ON state, Pol II is permitted to initiate transcription that is subject to a rate constant $k_{ini}$. When the promoter is in the OFF state, Pol II is unable to initiate transcription. The model also includes a transcription elongation step, which is assumed to be 100% processive, and whose time-scale depends on the gene length and the rate of elongation. The latter is assumed to be 1100 nucleotides/min, which is a value that has been experimentally determined in *Drosophila* (*Ardehali et al., 2009*).

In the model, transcriptional bursts have a characteristic size (number of transcripts per burst) and frequency (rate at which bursts occur). The average burst size is defined as $k_{ini}/k_{off}$, whereas the average burst frequency is defined as $(k_{on}^{-1} + k_{off}^{-1})^{-1}$ (*Dar et al., 2012*). We systematically and independently varied the parameters $k_{on}$, $k_{off}$ and $k_{ini}$ to tune the frequency and size of virtual bursts. For

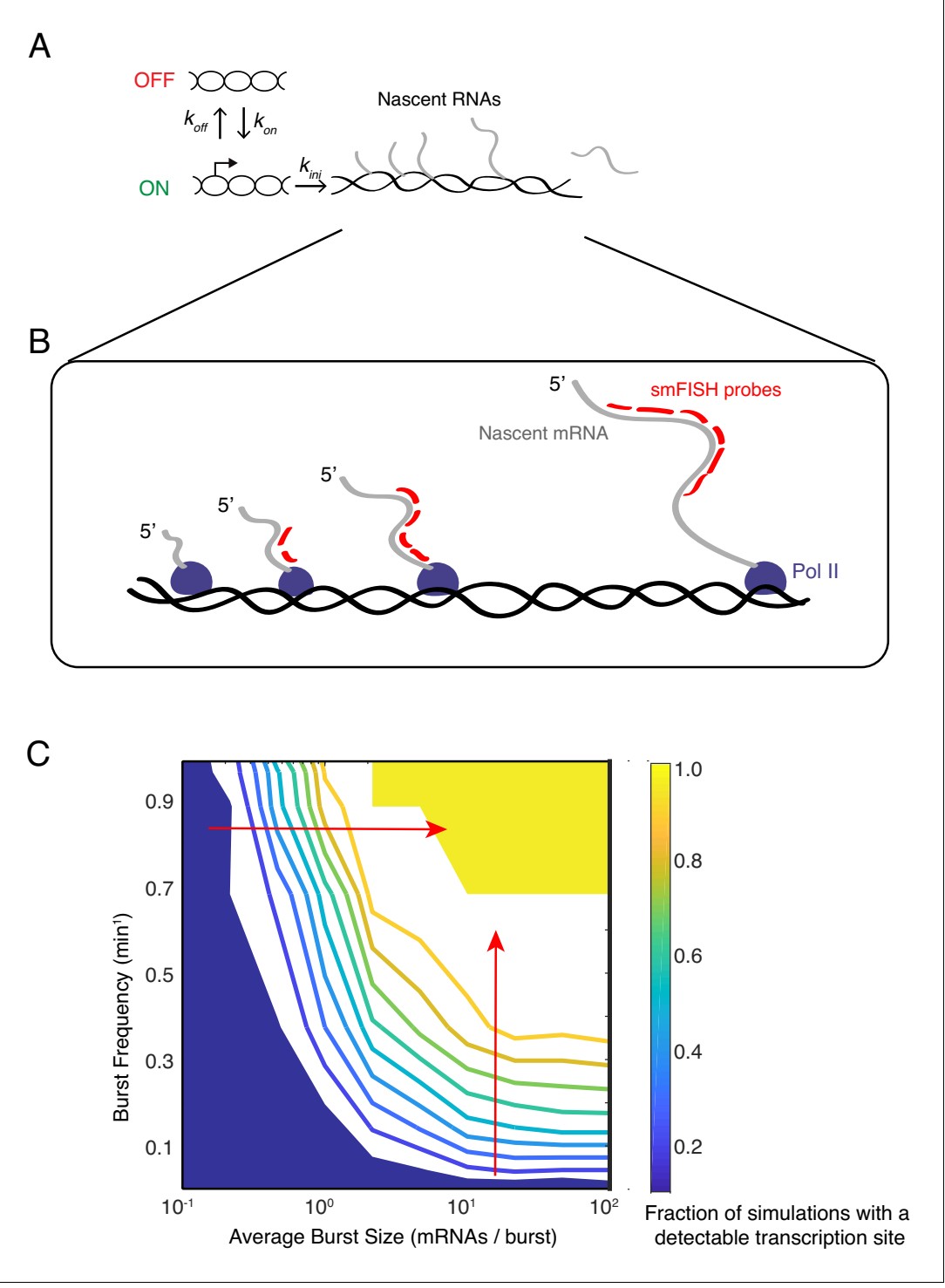

**Figure 4.** Modeling transcription sites using bursting dynamics. (**A**) Model framework showing the three rate parameters affecting transcription initiation. Two parameters affect the promoter state, while the third parameter only affects how many initiation events occur when the promoter is ON. (**B**) Pol II molecules in elongation mode are distributed along the transcription unit. If Pol II is upstream of the probe binding sites, the nascent transcript will not be detected. If Pol II is downstream, the nascent transcript will be detected as 100% signal. If Pol II is transcribing within the binding sites, the nascent transcript will be detected as a partial signal. These three different scenarios are explicitly accounted for in our mathematical model. For example in the simulation result shown here, four Pol II's are situated such that a total of 12 virtual probe-binding sites are present. Since each mRNA has six binding sites, it means that this simulated transcription site has 12/6 or 2 units of normalized signal.
*Figure 4 continued on next page*

*Figure 4 continued*

Applying our filter cutoff for identifying a transcription site as two or more units, this simulated site would be scored as a positive. (**C**) The phase diagram of transcription site detection as a function of burst size and frequency in the model. Both burst size and frequency impact the likelihood of detecting a transcription site. When burst size increases at low burst frequency, the likelihood of detecting a transcription site remains fairly constant. When burst size increases at high burst frequency (horizontal red arrow), the likelihood of detecting a transcription site is ultrasensitive to burst size. Likewise, when burst frequency increases at low burst size, the likelihood of detecting a transcription site remains fairly constant. When burst frequency increases at high burst size (vertical red arrow), the likelihood of detecting a transcription site is ultrasensitive to burst size. The phase diagram makes manifest that a range of combinations of burst frequency and size could explain observed transcription site frequency data.

each parameter set, we ran 1000 simulations of the model. To capture the stochastic nature of gene expression, reactions in the model were treated as probabilistic events, with the exception of the transcript elongation time.

To directly relate the results of model simulations to experimental data, we performed the following treatment of simulation data. First, we transformed output of each simulation to mimic the experimentally detected fluorescence at a single gene allele. Fluorescence intensity depends on how many probe-binding sites are present in nascent RNAs on a gene allele at a given time (*Figure 4B*). This varies with the number of elongating Pol II molecules on the allele, and the position of the probe-binding sites relative to the transcription start and stop sites. We normalized the output of simulated nascent RNAs by calculating the number of Pol II molecules upstream, within, and downstream of the binding region at the completion of a simulation. This normalization provided an approximation of fluorescence intensity from one gene allele. Second, we randomly paired two independent simulations to mimic the transcription site fluorescence of paired alleles within a nucleus. If simulated transcription site fluorescence fell below a cutoff of twice the fluorescence of a single RNA, we counted that simulation as having no 'detectable' transcription site. This mimicked the cutoff that was applied to experimental data for identifying a transcription site.

We then asked what combination of burst size and frequency could theoretically account for the observed frequency of finding cells with a transcription site (this ranged from 20% to 50% of cells). A phase diagram revealed that a broad range of burst size and frequency could explain our experimental observations (*Figure 4C*). Therefore, according to our model results, tuning burst frequency and/or size can produce the variable likelihood of detecting a transcription site by smFISH.

## Burst frequency is regulated by Dpp and Wg

We quantified the frequency of detecting a transcription site as a function of cell position within the wing pouch (*Figure 5A,B*). This frequency varied across the disc in a manner that was gene-specific. Strikingly, the spatial distributions of transcription site frequency strongly paralleled the mRNA number per cell for all genes (compare *Figure 5A,B* and *Figures 1G* and *2H*). To ensure that this was not an artifact of variable smFISH detection, we also quantified the frequency of detecting a transcription site for *sd*, which is uniformly expressed in the wing pouch. This frequency was constant across the disc and paralleled the *sd* mRNA number per cell (*Figure 2— figure supplement 1A,B*).

We further examined the relationship between mRNA number per cell and transcription site frequency (*Figure 5C,D*). Average mRNA number per cell and the likelihood of detecting a transcription site were linearly correlated with one another for all genes. The positive correlation confirms that Dpp and Wg regulate gene expression primarily through control of transcription initiation. Remarkably, the slopes of linear fits for three Dpp-responsive genes, *brk*, *omb*, and *salm*, were not significantly different from one another, and the slope for *dad* was similar to *brk* and *omb* but smaller than for *salm* (*Figure 5E*). This conserved linear relationship between gene transcription and mRNA number has several implications. It suggests that mRNA decay rates are not very different between these Dpp target genes since the slopes would be different from one another if decay rates varied. Moreover, since the slopes are constant over a broad range of mRNA output, it suggests that mRNA decay is not being actively regulated by Dpp.

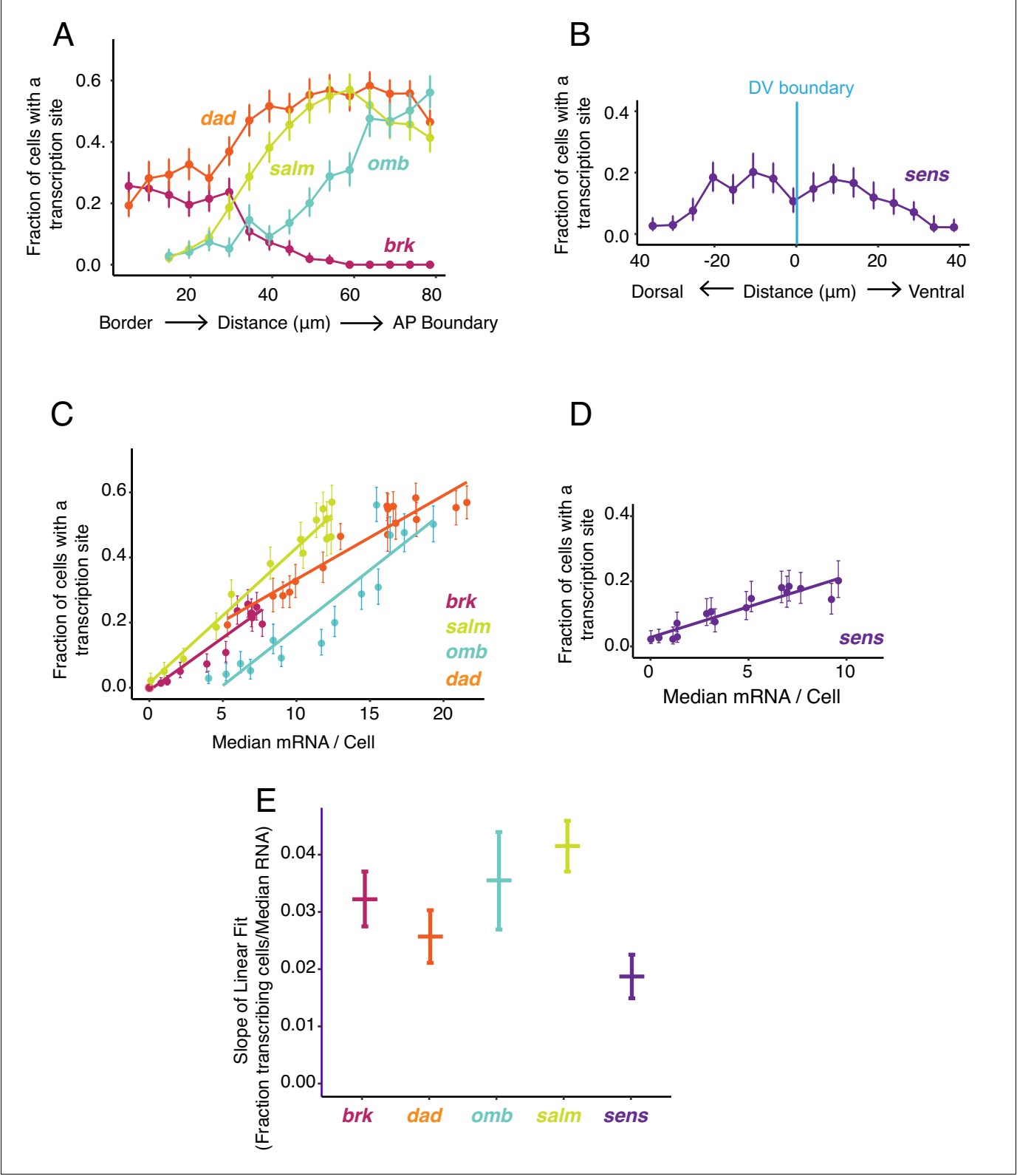

**Figure 5.** Transcription site detection correlates with mRNA number. (A,B) The probability of detecting a cell with a transcription site varies with the cell's location relative to the source of morphogen. Error bars are 95% bootstrapped confidence intervals. (A) Cells are binned according to their distance from the pouch border, and the fraction of cells in each bin with a transcription site are shown for each Dpp-responsive gene. (B) Cells are binned according to their distance from the DV boundary, and the fraction of cells in each bin with a transcription site is shown for the *sens* gene. (C,D)
*Figure 5 continued on next page*

*Figure 5 continued*

The probability of detecting a cell with a transcription site varies linearly with the number of mRNA molecules in the cell. Fitted lines are from linear regression. Error bars are 95% confidence intervals. (C) Cells are binned according to the number of mRNAs they contain, and the fraction of cells in each bin with a transcription site are shown for each Dpp-responsive gene. (D) Cells are binned according to the number of mRNAs they contain, and the fraction of cells in each bin with a transcription site is shown for the *sens* gene. (E) Linear regression analysis was performed on samples from C and D, shown is the slope with a parametric 95% confidence interval.

The likelihood of detecting a transcription site increases because either the promoter is spending more total time in the ON state or more RNAs are being transcribed while in the ON state. These properties are affected by burst size and burst frequency in different ways. We sought to determine whether burst size or frequency was being regulated. We did so by estimating the number of nascent RNAs at each transcription site, which was quantified as a multiple of the median pixel intensity of mature RNA spots (*Figure 3—figure supplement 1*). The average number of nascent RNAs per transcription site did not significantly vary between cells that were receiving different levels of Dpp and Wg signal (*Figure 6A,B*). This was observed for all genes, including the uniformly expressed *sd* gene (*Figure 2—figure supplement 1C*). Moreover, the average number of nascent RNAs per transcription site was also independent of the likelihood that transcription was occurring in a cell (*Figure 6C*). Therefore, the propensity for a cell to generate nascent transcripts does not correlate with the number of nascent transcripts.

To understand the causes of the relationship between these observed features, we turned to our mathematical model. We first considered whether modulation of transcription burst size by Wg and Dpp could explain our observations. We modulated burst size by systematically varying the $k_{ini}$ parameter, and from simulations, then calculated the number of nascent RNAs per transcription site and the transcription site detection frequency. There was a positive correlation between nascent RNA number in a transcription site and the probability of detecting a transcription site (*Figure 6D* and *Figure 6—figure supplement 1A*). This was observed across a wide range of fixed burst frequencies. When nascent RNA number was three or higher, the correlation with transcription site frequency was strongest. Moreover, when the probability of a transcription site was very low, nascent RNA number converged to a common value irrespective of burst frequency. None of these model predictions were observed in the experimental results with the target genes (*Figure 6C*). It suggests that transcription burst size is not strongly regulated by Dpp and Wg.

We then modulated burst frequency in the model by systematically varying $k_{on}$, and calculated the number of nascent RNAs per transcription site and the transcription site frequency. There was little change in nascent RNA number as transcription site frequency changed, even across a wide range of fixed burst sizes (*Figure 6E* and *Figure 6—figure supplement 1B*). The burst size appeared to determine what nascent RNA number value was held at a constant. Moreover, there was no convergence of nascent RNA number when the probability of a transcription site was very low, irrespective of burst size. All of these model predictions agree well with the experimental results (*Figure 6C*). This suggests that Dpp and Wg regulation of genes in the wing disc primarily occurs by modulation of transcriptional burst frequency.

## Discussion

Morphogens elicit different transcriptional outputs from target genes, depending on local concentration of the morphogen. The targets of Dpp signaling in the wing offer a well-studied example of this concept. Transcription of the gene *brk* is directly regulated by the Dpp effector protein Mothers-against-dpp (Mad) (*Minami et al., 1999*; *Moser and Campbell, 2005*). Mad, in complex with Medea and Schnurri, represses *brk* transcription (*Cai and Laughon, 2009*). This generates a gradient of Brk protein expression that is inverted to the Dpp gradient. In turn, the level of Brk protein is instrumental in repressing the expression of *omb* and *salm*, which are induced by Dpp (*Campbell and Tomlinson, 1999*). Thus, opposing gradients of activation and repression determine the expression domains of *omb* and *salm*. Since *omb* is less sensitive to Brk repression than *salm*, its expression domain is broader. *salm* transcription is directly activated by Dpp without participation of Schnurri (*Moser and Campbell, 2005*). Curiously, *omb* transcription does not directly depend on

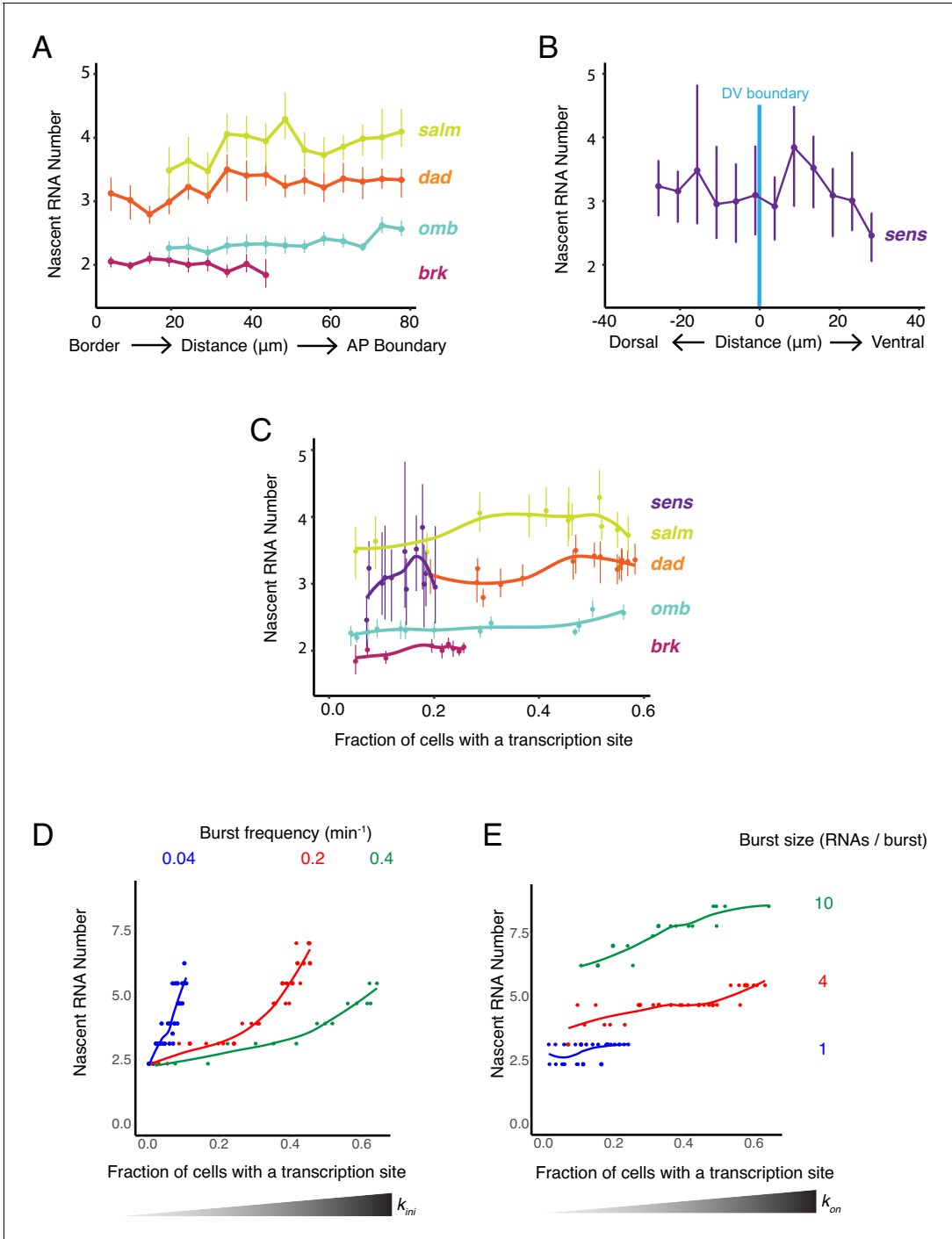

**Figure 6.** Burst frequency is regulated by Dpp and Wg. (**A,B**) The average number of nascent RNAs in a transcription site does not vary with the cell's location relative to the source of morphogen. Error bars are bootstrapped 95% confidence intervals. (**A**) Cells are binned according to their distance from the pouch border, and the average number of nascent RNAs per site in each bin are shown for each Dpp-responsive gene. (**B**) Cells are binned according to their distance from the DV boundary, and the average number of nascent RNAs per site in each bin is shown for the sens gene. (**C**) The average number of nascent RNAs in a transcription site does not vary with the probability of detecting a cell with a transcription site. Error bars are 95% confidence intervals. (**D,E**) Model predictions of the relationship between average number of nascent RNAs in a transcription site and the probability of detecting a site for the *dad* gene. (**D**) Simulations are performed where the rate parameter $k_{ini}$ has been systematically varied so as to modulate burst size alone. Resulting values for nascent RNA number and fraction of cells with a site are shown. Each datapoint is the average of 1000 simulations. Simulations are repeated for three different values of $k_{on}$ to specifically set the burst frequency to 0.04, 0.2 and 0.4 min$^{-1}$. (**E**) Simulations are performed where the rate parameter $k_{on}$ has been systematically varied so that burst frequency alone is variable. Resulting values for nascent RNA number and

*Figure 6 continued on next page*

*Figure 6 continued*

fraction of cells with a site are shown. Each datapoint is the average of 1000 simulations. Simulations are repeated for three different values of $k_{ini}$ to specifically set the burst size to 1, 4 and 20.

The online version of this article includes the following figure supplement(s) for figure 6:

**Figure supplement 1.** Modeling the relationship between average number of nascent RNAs in a transcription site and the probability of detecting a site for the *brk, omb, salm,* and *sens* genes.

Dpp signaling, and its transcriptional activation is brought about by unknown factors (*Sivasankaran et al., 2000*).

Given the diverse molecular mechanisms by which genes such as *omb*, *brk*, and *salm* are regulated, it is illuminating that regulation of transcription burst frequency occurs for all of them. In the two-state view of promoter kinetics, the on-rate, $k_{on}$, then is the most likely rate constant being regulated since it specifically affects burst frequency alone (*Dar et al., 2012*). It determines the average rate at which a promoter will switch from its OFF to its ON state. When a promoter is in the OFF state, the next burst will only occur when it switches ON, which is controlled by $k_{on}$ and not $k_{off}$. When a promoter is in the ON state, the size of its burst depends on when it switches OFF, which is controlled by $k_{off}$ and not $k_{on}$ (*Dar et al., 2012*). However, $k_{off}$ also affects burst frequency because the longer a promoter is ON, the longer the time it takes before a new burst can occur. If Dpp regulates $k_{off}$, then we would have seen modulation of both size and frequency of bursts. However, burst size appears to be independent of Dpp signaling.

If $k_{on}$ is the kinetic rate constant under regulation for all of these genes, how does this occur given such diverse enhancer architectures and transcription factor inputs? It has been found that burst frequency correlates with enhancer strength and enhancer-promoter contact, suggesting that $k_{on}$ is potentiated by enhancer-promoter contact and is mediated by transcription factor binding to DNA (*Bartman et al., 2016*; *Bothma et al., 2014*; *Chen et al., 2019*; *Fukaya et al., 2016*; *Larsson et al., 2019*). This suggests that occupancy of Dpp effectors on target enhancers varies the $k_{on}$ rate for their linked promoters, and this modulation is negative for repressors such as Brk and positive for activators such as Mad.

Burst frequency regulation is also observed for developmental genes in the embryo (*Bothma et al., 2014*; *Chen et al., 2019*; *Fukaya et al., 2016*; *Garcia et al., 2013*; *Holloway and Spirov, 2017*; *Little et al., 2013*; *Xu et al., 2015*). Thus, a common mechanism to regulate patterned gene expression is by control of burst frequency. However, burst size can also be regulated by cell-cell signaling, as is the case for Notch target genes in the *Drosophila* embryo (*Falo-Sanjuan et al., 2019*). Moreover, *eve* gene expression in the embryo is regulated by transcription factors that modulate burst frequency, plus there is an orthogonal mechanism that controls the window of time over which a nucleus can transcribe the *eve* gene (*Lammers et al., 2020*). This distinct mechanism appears to be regulated by repressors, perhaps acting on nucleosome organization. Modeling of various embryonic genes suggests that they transition through several intermediate transcriptionally-silent states before their transcription can begin (*Desponds et al., 2016*; *Dufourt et al., 2018*; *Eck et al., 2020*). Chromatin remodeling factors appear to modulate these transitions (*Eck et al., 2020*). Although a two-state model explains much of our experimental results, likely there are other factors that also help determine the expression domains of Dpp-responsive genes.

Our results challenge the view that *salm* and *omb* expression domains have sharp boundaries due to transcription thresholds set by Brk and Dpp. We find that *omb* and *salm* mRNA numbers per cell drop gradually with distance from the source of Dpp (*Figure 2H*). As well, their gradients in mRNA number are inversely correlated with the gradient in *brk* mRNA number. *Salm* has relatively constant mRNA number in cells near the AP boundary, and those numbers gradually diminish in cells located more laterally. A similar pattern is seen with *omb*, except the domain with constant *omb* mRNA number is smaller than for *salm*. However, the *salm* and *omb* enhancer trap reporters as well as anti-Salm immunohistochemistry have reported expression domains with sharp boundaries (*Mayer et al., 2013*). Possibly, the discrepancy hints at some threshold of mRNA expression below which protein output drops sharply. It is also possible that the previously characterized expression domains for

*salm* and *omb* were distorted by non-linear detection of antibodies that recognize Salm and the protein product of lacZ, β-galactosidase.

# Materials and methods

## Key resources table

| Reagent type (species) or resource | Designation | Source or reference | Identifiers | Additional information |
|---|---|---|---|---|
| Gene (*Drosophila melanogaster*) | *white*[1118] | Bloomington *Drosophila* Stock Center | *BDSC*: 3605 *Flybase*: FBst0003605 RRID:BDSC_3605 | |
| Gene (*Drosophila melanogaster*) | *sens*[E1] | **Nolo et al., 2000** | *Flybase*: FBal0098024 | From Hugo Bellen |
| Genetic reagent (*Drosophila melanogaster*) | *sfGFP-sens [VK37]* | **Venken et al., 2006**. From Hugo Bellen | | Pacman construct containing *sens* gene with N-terminal 3xFlag-TEV-StrepII-sfGFP-FlAsH fusion tag inserted at 22A3 (VK37) |
| Genetic reagent (*Drosophila melanogaster*) | *dad-GFP [VK37]* | Bloomington *Drosophila* Stock Center | *BDSC*: 81273 *Flybase*: FBti0150281 RRID:BDSC_81273 | *y w; PBac{y[+mDint2] w[+mC]=Dad GFP.FLAG} inserted at 22A3 (VK37)* |
| Genetic reagent (*Drosophila melanogaster*) | *brk-GFP [VK33]* | Bloomington *Drosophila* Stock Center | *BDSC*: 38629 *Flybase*: FBti0147730 RRID:BDSC_38629 | *w[1118]; PBac{y[+mDint2] w[+mC]=brk GFP.FPTB} inserted at 65B2 (VK33)* |
| Sequence-based reagent | *GFP* hybridization oligo probes | Biosearch Technologies | Custom probe set | Set of oligos with 3' modification mdC (TEG-Amino). Sequence of all oligos is in **Supplementary file 1** |
| Sequence-based reagent | *sens* hybridization oligo probes | Biosearch Technologies | Custom probe set | Set of oligos with 3' modification mdC(TEG-Amino). Sequence of all oligos is in **Supplementary file 1** |
| Sequence-based reagent | *salm* hybridization oligo probes | IDT | Custom probe set | Set of oligos. Sequence of all oligos is in **Supplementary file 1** |
| Sequence-based reagent | *omb* hybridization oligo probes | IDT | Custom probe set | Set of oligos. Sequence of all oligos is in **Supplementary file 1** |
| Sequence-based reagent | *sd* hybridization oligo probes | IDT | Custom probe set | Set of oligos. Sequence of all oligos is in **Supplementary file 1** |
| Sequence-based reagent | *omb* intron hybridization oligo probes | IDT | Custom probe set | Set of oligos. Sequence of all oligos is in **Supplementary file 1** |
| Sequence-based reagent | *omb* 5' exon hybridization oligo probes | IDT | Custom probe set | Set of oligos. Sequence of all oligos is in **Supplementary file 1** |
| Chemical compound, drug | NHS-ester ATTO 633 dye | Sigma | #01464 | |
| Chemical compound, drug | NHS-ester ATTO 565 dye | Sigma | #72464 | |
| Chemical compound, drug | amino-11-ddUTP | Lumiprobe | A5040 | |
| Chemical compound, drug | Paraformaldehyde (powder) | Polysciences | 00380–1 | |
| Chemical compound, drug | Triton X-100 | Sigma Aldrich | T9284-500ML | |
| Chemical compound, drug | VectaShield | Vector Labs | H-1000 | |

*Continued on next page*

*Continued*

| Reagent type (species) or resource | Designation | Source or reference | Identifiers | Additional information |
|---|---|---|---|---|
| Chemical compound, drug | 4',6-diamidino-2-phenylindole (DAPI) | Life Technologies | D1306 | |
| Chemical compound, drug | salmon sperm single stranded DNA | Invitrogen | #15632 | |
| Chemical compound, drug | vanadyl ribonucleoside | New England Biolabs | #S14025 | |
| Software, algorithm | MATLAB pipeline to process raw smFISH images with no prior preprocessing | This paper | | https://github.com/elifesciences-publications/smfish_pipeline |
| Other | Graces' Insect Medium | Sigma | #69771 | Growth medium for organ culture |

## Experimental model and subject details

For all experiments, *Drosophila melanogaster* was raised using standard lab conditions and food. Stocks were either obtained from the Bloomington Stock Center, from listed labs, or were derived in our laboratory (RWC). A list of all stocks and transgenics used in this study is in the Key Resources Table. The sample sizes were not computed when the study was designed. Sample sizes were determined such that >6,000 cells were measured for each genotype. All *Drosophila* were raised at room temperature and grown on standard molasses- cornmeal food. The *sfGFP-sens* transgenic line was used as described in *Cassidy et al., 2013*. Experiments were performed on *dad-GFP* and *brk-GFP* transgenes obtained from Bloomington *Drosophila* Stock Center (stocks 81273 and 38629, respectively). For all transgenic experiments, smFISH was performed on homozygous individuals. Experiments were performed on endogenous *omb* and *salm* in $w^{1118}$ individuals. There was no exclusion of any data or subjects.

## Method details

smFISH Probe Design and Preparation smFISH oligonucleotide probes were designed using Stellaris Probe Designer (Biosearch Technologies). Probes sets contain between 45 and 48 non-overlapping 20-nucleotide oligos. A full list of all probe sets is provided in *Supplementary file 1*. Anti-GFP probes were prepared by conjugating NHS-ester ATTO 633 dye (Sigma 01464) to the 3' end of each oligonucleotide. Anti-Sens probes were prepared by conjugating NHS-ester ATTO 565 dye (Sigma 72464) to the 3' end of each oligonucleotide. These oligos bear a mdC(TEG-Amino) 3' modification to allow conjugation, and were obtained from Biosearch Technologies. Conjugation and purification was performed as described (*Little and Gregor, 2018*). All other probe sets were prepared using the enzymatic conjugation protocol as described (*Gaspar et al., 2017*). Briefly, amino-11-ddUTP (Lumiprobe) was conjugated to NHS-ester ATTO 633. Terminal deoxynucleotidyl transferase (New England Biolabs) was then used to conjugate ATTO 633-ddUTP to the 3' ends of oligonucleotides that had been purchased from IDT. After enzymatic conjugation, oligos were purified from free ATTO 633-ddUTP using G-25 spin columns (GE Illustra) according to manufacturer's instructions. Final concentration of oligonucleotide was 33 μM in water. Probes were stored at −20°C, protected from light, until use.

## smFISH

Wing discs were dissected from wandering 3rd instar larva in cold phosphate buffered saline (PBS) and immediately fixed in 0.1% (w/v) paraformaldehyde/PBS for 15 min at room temperature. Discs were then fixed for 30 min in methanol at room temperature. Discs were transferred to hybridization buffer (10% w/v dextran sulfate, 4X SSC, 0.01% w/v salmon sperm ssDNA (Invitrogen 15632), 1% v/v vanadyl ribonucleoside (NEB S14025), 0.2 mg/mL BSA, 0.1% v/v Tween-20). Oligo probes were added to a 1.5 μM final concentration in the hybridization buffer, and hybridization was performed for 1 hr at 62° C. After hybridization, discs were washed once for 5 min at 62° C in wash buffer (4X SSC, 0.1% v/v Tween-20). Discs were then incubated with 2.5 μg/mL 4',6-diamidino-2-phenylindole

(DAPI) (Invitrogen) in PBS + 0.1% Tween-20 for 5 min at room temperature. Discs were washed with PBS + 0.1% Tween-20 and transferred to Vectashield (Vector Labs) for mounting. Discs were mounted in 15 µl of Vectashield on glass microscope slides using an 18 × 18 mm No. one coverslip (Zeiss). For eye imaginal discs, discs were dissected from late 3rd instar larva in cold PBS with brain and mouth hooks attached, then smFISH was performed as described. Immediately prior to mounting, brain and mouth hooks were removed from eye discs and discarded.

## Actinomycin D treatment

Wing discs were dissected in room temperature Graces' Insect Medium (Sigma 69771) supplemented with 1X Pen-Strep (Gibco 15140–122) and 5 mM Bis-Tris (Sigma B4429). Half of the total dissected discs were transferred to 24-well tissue culture dishes containing this prepared medium + 5 µg/mL Actinomycin D, and half were transferred to untreated controls containing culture medium + 1:1000 (v/v) DMSO. Discs were incubated with gentle shaking for 30 min at room temperature, protected from light, before being washed with fresh culture media, and 1X PBS. SmFISH was then performed as described.

## Image acquisition

3D image stacks were collected on a Leica SP8 scanning confocal microscope, using a pinhole size of 1 Airy unit and a 63X oil immersion (NA 1.4) objective. Approximately, 35 optical sections were collected per sample, with each section 700 nm thick. Sections were spaced 345 nm apart. DAPI, ATTO 565, and ATTO 633 were excited by 405, 555, and 630 nm lasers, respectively. ATTO dye fluorescence was collected using a HyD detector on photon counting mode and a scanning speed of 200 Hz, with 16X line accumulation. DAPI fluorescence was collected using a PMT detector using 8X line averaging. Pixel intensity values are 12-bit, and x-y pixel sizes are 76 nm. We modeled each z-section like a plane of width 345 nm for analysis, but in reality the edges of the z point spread function (PSF) overlap between sections. Since the PSF resembles a Gaussian distribution, most of the light is coming from the center of that distribution. Therefore, overlap is needed between sections to ensure equivalent sampling of the entire specimen.

## Image processing

Raw smFISH images were processed using a custom Matlab pipeline with no prior preprocessing. Our pipeline is available at https://github.com/bakkerra/smfish_pipeline (*Bakker et al., 2020*; copy archived at https://github.com/elifesciences-publications/smfish_pipeline). The pipeline consists of several modules.

### Selection of mRNA Segmentation Threshold

RNA segmentation is performed by applying a cutoff intensity value to a stack of optical sections, and transforming all pixels above the cutoff to white and pixels below the cutoff to black. Diffraction-limited fluorescent spots captured with a 63X objective at 633 nm wavelength are estimated to be approximately 600 nm in diameter (*Lipson et al., 1995*). This corresponds to a diameter of 8 x-y pixels in our images. Therefore, we classify a 2D object in each transformed section when $\geq$ 8 white pixels are connected with one another.

It is important to select a cutoff where true RNA fluorescent spots are identified as 2D objects in a section, and background is not. Therefore, a broad range of cutoff values is systematically applied to an image stack, and 2D object number is summed for each cutoff value. The distribution of 2D object number exhibits a plateau across a range of cutoff values (*Figure 1—figure supplement 1C*), and thresholds applied at this plateau accurately identify true RNA spots. To demonstrate this, we manually curated 347 RNA spots from sub-regions of four independent image stacks, and found that when a cutoff is selected within the plateau, the number of 2D objects identified by threshold segmentation is no more than +/- 5% different than the ground truth. Furthermore, the centroids of identified objects have an average displacement of only two pixels from the manually identified centroids. Therefore, this plateau, a regime of relative insensitivity to user-specified hyperparameters, is an appropriate threshold for accurate segmentation of RNA spots.

The position of the plateau varies from image stack to image stack. Therefore, for each image stack, a range of cutoffs is tested, and a cutoff is selected within the plateau to perform

segmentation. As a result, each image stack has a unique threshold, allowing robust segmentation of spots despite variation in raw fluorescence between image stacks. In practice, replicate samples from the same experiment captured in the same imaging session did not require thresholds for segmentation more than 15 pixel intensity units apart. If image stacks did not show an identifiable plateau, the signal-to-background of that sample was determined to be insufficient and it was not used for analysis. The smFISH protocol and imaging is robust enough that in our hands, this occurs in less than 10% of image stacks collected. Once a threshold is selected, the following properties of each 2D object are recorded: x-y centroid position, z-section, and a list of the connected pixels.

## Connecting 2D Objects into 3D mRNA Spots

As each z-section is 340 nm in depth, it is assumed that genuine diffraction-limited RNA spots will appear in 2 or three consecutive z-sections, depending on the spot's position along the z-axis. Therefore, a 2D object must satisfy two criteria in order to be counted as an RNA spot: 1) A 2D object must be linked to one or more 2D objects in at least one neighboring z-section. Linkage is defined when the centroids of all objects are within a diffraction-limited radius of 4 pixels from one another in the x-y plane; 2) A 2D object must be larger (contain more pixels) than linked objects in neighboring z-sections. This criterion prevents RNA fluorescence spots from being counted in multiple z-sections. A candidate that satisfies these criteria is recorded as an mRNA spot, and only the largest 2D object is recorded.

The pipeline allows the images to be overlaid with markers indicating recorded spots so that each image stack can be manually inspected for any significant errors or inconsistencies. The most common problem detected at this stage results from images taken of discs that were 'drifting' or moving significantly, which can cause a large number of identified spots to be filtered out during processing for not meeting criterion 1. Excessive bleaching across the stack can also cause clear inconsistencies. In this study, such problems were rare enough that any sample experiencing these problems was considered to have failed quality control and was simply not included for further analysis.

Intensity measurements are recorded from a circle of pixels of radius four about the centroid of each recorded RNA spot. By keeping the area of each intensity measurement fixed, we uncouple user-generated variation in selection of segmentation thresholds from spot intensity measurements. A 2D circle was used instead of a 3D sphere to extract intensity measurements because the spots only appear in 2 or 3 z-sections. This makes their 3D geometry variable from spot to spot, and they cannot be consistently described using a sphere or ellipse.

## Segmentation of Transcription Sites

In our images, transcription sites tend to contain pixels that are many times brighter than mature RNA spots. As a result, the brightest transcription sites are frequently misidentified during segmentation of mature RNA spots because the second criterion for spot identification only records the largest object within the diffraction limit in z. For transcription sites, this is not always the brightest plane. Therefore, we segment transcription sites independent of mature RNAs using a higher cutoff. The objective in cutoff selection for transcription sites is to select one that includes objects with a total fluorescence intensity of twice the average mature RNA, and excludes mature RNA spots. We define the 'average' intensity for a spot containing a single mRNA to be the median of the distribution of all identified mature RNA spot objects. We empirically determined that merely doubling the cutoff for segmentation does not achieve this, because mature mRNAs may contain a few pixels above the cutoff, enough to still be identified as objects and included in analysis. Therefore, we use a threshold cutoff by multiplying the median mature RNA intensity by a factor of 2.5 (*Figure 3—figure supplement 1A*).

To test the accuracy of this segmentation procedure, we manually inspected three particularly RNA-dense regions in independent images where automated segmentation found a total of 103 transcription sites and 4066 mature RNAs. We determined that only 7 of 4066 mature RNAs were misidentified as transcription sites, and found no examples of transcription sites that had been missed by automated segmentation. After identification, object intensity measurements are recorded from a circle of pixels of radius 4 (the diffraction limit) about the centroid of each identified transcription site (*Figure 3—figure supplement 1B*). The average transcription site threshold

selected for replicates in a dataset show no correlation with the average intensity of transcription sites in that dataset (*Figure 3—figure supplement 1D*). Therefore, the differences in transcription site intensity between genes cannot be explained merely by differences in threshold selection or variability in image fluorescence between datasets.

## Estimation of Nascent RNA Number per Transcription Site

The intensity measurement of each identified transcription site in an image stack is divided by the median intensity of identified mature RNAs in that sample (*Figure 3—figure supplement 1C*). This serves two purposes. First, it serves to normalize these measurements within each sample so transcription site intensity measurements can be pooled across replicates without the effects of image-to- image variability in fluorescence. Secondly, each transcription site object is presumed to be the sum of intensities of multiple nascent RNA molecules elongating at the transcription site. By dividing each transcription site intensity by the average intensity of a single RNA, we obtain an estimate of the number of nascent RNAs present at the transcription site. Because some transcripts are partially elongated, this number cannot be completely accurate, and we attempt to compensate for this in our computational model when interpreting results.

## Nuclei Segmentation

DAPI fluorescence images are output as labeled 16-bit images, where each nuclear object corresponds to a 'level' in the 16-bit image. These images are input to a nuclei segmentation pipeline, which flattens the images to white nuclei objects and black background. Nuclei images are segmented in 2D using the NucleAIzer platform maskRCNN Network, trained as described in *Hollandi et al., 2020*. We trained the neural network with an expected nuclear radius of 32 pixels (*Figure 1—figure supplement 2D*). To ascertain the accuracy of segmentation, we compared results to manually labeled nuclei in four randomly selected disc images. The automated method identified at least 85% of nuclei objects identified manually for each image.

The segmented black and white images are then processed using a custom Matlab script in order to join overlapping 2D objects into 3D. Each nucleus object in each z-slice is assigned an identity index. For each object in the first z-slice, the object with the highest number of overlapping pixels in the next z-slice is identified, and this object's identity index is altered to be identical to its overlap partner. This proceeds through the entire z-stack of images, creating objects that resemble 'pancake stacks' of linked 2D objects in 3D (*Figure 1—figure supplement 2E*). The 3D-centroid and list of included pixels of these new objects is then recorded. Objects not incorporated into a 3-D object are disregarded.

## Generation of Voronoi Diagrams

A Voronoi tessellation is built from a grid of points in either 2D or 3D. In our case, each point is the centroid of a segmented 3D nucleus. The Voronoi cells delineate regions consisting of all voxels that are closer to that centroid than to any other centroid (*Voronoi, 1908*). The boundaries between Voronoi cells represent points that are equidistant between two centroids. These are taken to represent virtual cell boundaries. It is important to be clear that the Voronoi cells do not accurately describe the pseudostratified epithelial nature of wing disc cells. However, note that 3D segmentation of pseudostratified epithelial cells is still something no one working in any system has achieved.

The 3D Voronoi tessellation used a polytope-bounded Voronoi diagram available for Matlab, which uses the DeLaunay triangulation to calculate the Voronoi diagram (*Park, 2020*). The result of this tessellation is a list of 3D vertices of each Voronoi 'cell' in space, which is recorded along with the associated nuclear centroid (*Figure 1—figure supplement 2F*).

## Assignment of RNA to cells

3D Voronoi tessellation is a way to democratically assign mature transcripts to cells based on vicinity to the nucleus. Clearly we get the transcript assignment incorrect at a local level. However, being the most democratic approach, the trends of mRNAs/cell assigned to 100's of cell across the wing disc are trustworthy. The same logic has been used by others assigning mRNA transcripts to early embryonic nuclei when cell boundaries are unseen (*Little et al., 2013*).

To assign spot objects to cells, a 3D convex hull of the each Voronoi cell is constructed from the vertices data for that cell. An entire set of image points, either the mRNA or transcription spot centroids, are tested to determine whether they fall inside or outside of each hull (*Figure 1—figure supplement 2E*). This is performed using a Matlab function called inhull, which uses dot products to shorten calculation times (*D'Errico, 2020*). Spots that fall inside a given cell's Voronoi hull are assigned to that cell's nuclear centroid, and the number of assigned spots, as well as their centroid and z- plane information are recorded. This is then repeated for every Voronoi cell in the image stack. The final result is a list of cells, their nuclear centroids, the total number of RNA spots assigned, and a list of each assigned spot's centroids.

## Data Analysis Binning of data

Each disc is imaged with the DV boundary located at the y-coordinate midline of the image. Therefore the x-coordinate of the image corresponds to position along the disc's AP axis, and the y-coordinate corresponds to position along the DV axis. In order to analyze data across developmental axes, each image is divided into spatial bins of 64 pixels each, approximately equal to the diameter of one cell nucleus. RNA spots are assigned to a bin according to the position of their associated nuclear centroid.

## Sample size and replicates

We analyzed image stacks from three independent discs for each experiment. Each image stack contains approximately 1,700 identified nuclei. Therefore, the total sample size is approximately 5,000 cells per experiment. Similar trends in RNA and transcription spots feature are observed in each disc individually, and hence, the analysis is not distorted by artifacts in pooling and cell segmentation (*Figure 3—figure supplement 2*).

## Alignment of replicates along developmental axes

While each disc is imaged roughly in the same region, there is not an unambiguous landmark that precisely registers different disc images with one another. To pool data across space as accurately as possible, we register discs to each other based on their mRNA spot distributions over space. For each image data set, the number of RNAs per spatial bin is summed, and the distributions across bins are compared. Bins are then manually registered such that the distribution profiles of the three datasets line up with one another (*Figure 3—figure supplement 2A–E*). The overlapping bins from the three datasets are then assigned to a pooled bin. Pooling includes the nuclei centroids as well as the transcription and RNA spots. This is repeated for all bins.

## Calculations

Median mature mRNAs per cell is calculated from total number of mature mRNA spots for each cell within a spatial bin of pooled data. As the distribution of mRNAs per cell is not normally distributed and has a long tail, we ascertained that the median was a more robust descriptor of the 'center' of the distribution than mean. *Median nascent RNAs per cell* is calculated from normalized intensity measurements for each transcription spot within a spatial bin of pooled data. All nascent RNA spots are included. As the distribution of RNA per cell is not normally distributed and has a long tail, we ascertained that the median was a more robust descriptor of the 'center' of the distribution than mean. Because the number of transcription sites varies over space, sample sizes vary for calculating median nascent RNAs per cell. For bins where fewer than 5% of cells contain a transcription site, median nascent RNAs per cell was not calculated, as the sample size was determined to be too small (<15). *Fraction of cells with a transcription site* is calculated by dividing the number of cells in a pooled spatial bin with at least one transcription site assigned to them by the total number of cells in that spatial bin. *Fano factor* is calculated for each spatial bin by dividing the variance in the mRNA per cell distribution by the mean mRNA per cell for all cells assigned to that pooled spatial bin.

## Statistics

Linear models are produced by unweighted least squares linear regression. LOESS fits are performed using the loess fitter in R, with an optimized span to minimize residuals. Confidence intervals are calculated by bootstrap resampling analysis using the bias-corrected and accelerated method.

We resample data within each bin of pooled data and calculate the statistic of interest 10,000 times. The mean value of the statistic and a 95% confidence interval are calculated from these resampled values.

## Stochastic simulation model

We model the various steps of gene expression, based on central dogma, as linear first order reactions. To simulate the stochastic nature of reactions, we implement the model as a Markov process using Gillespie's Stochastic Simulation Algorithm (*Gillespie, 1977*). Simple Markov processes can be analyzed using a chemical master equation to provide a full probability distribution of states as they evolve through time. The master equation defining our gene expression Markov process is as follows:

$$
\begin{aligned}
\frac{\partial P(N_m, N_g, t)}{\partial t} = \ & K_{ini}\left[(N_m - 1)P(N_m - 1, N_g, t) - P(N_m, N_g, t)\right] \\
& + K_{deg}\left[(N_m + 1)P(N_m + 1, N_g, t) - N_m P(N_m, N_g, t)\right] \\
& + K_{on}\left[\left((N_g - N_{gtot})\right)P(N_m, N_g - 1, t) - (N_g - N_{gtot})P(N_m, N_g, t)\right] \\
& + K_{off}\left[(N_g + 1)P(N_m, N_g + 1, t) - N_g P(N_m, N_g, t)\right]
\end{aligned}
$$

where $N_m$, $N_g$, and $t$ are defined as the number of RNA molecules present, as the number of transcriptionally active gene copies, and simulation time, respectively. $N_{gtot}$ is defined as the total number of gene copies present, and thus is the maximum number of active gene copies that can exist in the simulation. $K_{ini}$, $k_{deg}$, $k_{on}$, and $k_{off}$ are rate constants defining the rates of transcription initiation, RNA degradation, promoter state switching from off to on, and promoter state switching from on to off, respectively.

As the Markov process gets more complex, the master equation can become too complicated to solve. Gillespie's Algorithm is a statistically exact method that generates a probability distribution identical to the solution of the corresponding master equation given that a large number of simulations are realized. A brief description of how the Gillespie simulation produces each probability distribution is as follows:

1. We initialize all simulations to start with no mRNA molecules and promoter state is set to OFF.
2. For each event $i$ in the simulation, a total rate $r_{tot}$ is calculated by summing all $r_i$ reaction rate constants in the model, given the current promoter state and the total number of mRNA molecules present.
3. A time-step $\tau$ is generated from an exponential probability distribution with mean $1/r_{tot}$. This $\tau$ is the time interval between the current event and the next event.
4. Each event $i$ is selected from the list of reaction steps in the model available at that time (promoter switching, transcription initiation, mRNA decay). The probability a reaction step is selected is equal to $r_i/r_{tot}$. An event is selected at random given these probabilities. For each event, the following actions are taken:
   - Promoter switches to ON: Promoter is now in ON state, transcription initiation is now included in $r_{tot}$,
   - Promoter switches to OFF: Promoter is now in OFF state, transcription initiation is no longer included in $r_{tot}$.
   - Transcription Initiation: Number of mature mRNA molecules is increased by 1.
   - RNA degradation: Number of mature mRNA molecules is decreased by 1.
5. Simulation time is updated as $t + \tau$ where $t$ is the total time elapsed in the simulation.

Each simulation is run for 10,000 iterative events to approximate steady-state conditions, at the end of which the number of mRNA molecules present in the simulation is recorded. Independent simulations are then randomly paired to mimic the two alleles within a cell, and the sum of mRNA numbers is recorded as the mRNA output per cell. A minimum of 1000 simulation pairs are generated for each set of rate parameter values.

The RNA decay parameter $k_{deg}$ is fixed at 0.04/min for all simulations, as this rate had been experimentally determined for *sens* mRNA (*Giri et al., 2020*). The transcriptional rate parameters are varied in accordance with the specific hypothesis being tested. We constrain them loosely to be within an order of magnitude of reported values for these rates from the literature (*Milo et al., 2010*). We also constrain these rates so as to produce steady state mRNA numbers similar to experimental data.

- $k_{ini}$ is varied from 0.2 to 60/min
- $k_{on}$ is varied from 0.008 to 38/min
- $k_{off}$ is varied from 0.016 to 20/min

To perform a parameter sweep, we vary the relevant parameter across the defined range. Each rate parameter value in the sweep is used to make 1000 paired simulations as described above.

### Nascent Transcripts

Thus far we have described how model simulations generate in silico data for mature mRNA numbers. We also use the same simulations to approximate the number of nascent RNAs per gene. After 10,000 iterative events are completed in a simulation, the number of nascent RNAs is counted. A single nascent RNA is counted if a single transcription initiation event has occurred within an interval of time ($\tau_{elong}$) equal to the time it is estimated that RNA polymerase takes to elongate from the binding site for the 5'-most oligo probe to the 3' end of the RNA. To calculate $\tau_{elong}$ for each gene, we divide the number of nucleotides from 5' probe-binding site to 3' end by the transcription elongation rate. This rate is assumed to be 1100 nucleotides/min, as experimentally determined (*Ardehali et al., 2009*).

| Gene | $\tau_{elong}$ (min) |
|---|---|
| *brk* | 1.35 |
| *dad* | 2.05 |
| *sens* | 5.15 |
| *salm* | 5.30 |
| *omb* | 3.05 |

We weight the count of nascent RNAs in a simulation to mimic the fluorescence output from these nascent RNAs if they are hybridized to probes. We define $\tau_{probe}$ to be the time interval for RNA polymerase to elongate from the 5'-most probe-binding site to the 3'-most probe-binding site. If a nascent RNA had been initiated in a time less than $\tau_{probe}$, then we weight the counting of that nascent RNA as 0.5 rather than 1. We do this because the probe-binding region of the nascent RNA is partially transcribed at this point. For simplicity, the exact locations of probes and RNA polymerase are not taken into account to calculate the weighting, and instead we assign the overall probability of fluorescence for an ensemble of such partially transcribed RNAs. If a nascent RNA had been initiated in a time greater than or equal to $\tau_{probe}$ and less than $\tau_{elong}$, then we weight the counting of that nascent RNA as 1. These RNAs are assumed to produce 100% of the fluorescence of a mature RNA spot, since all probe-binding sites are transcribed at this point.

We randomly pair two simulations and sum the number of weighted nascent transcripts. This mimics the experimental conditions where the two gene alleles are physically paired and thus their nascent RNAs are co-localized in space. We collate 1000 paired simulations for each parameter set and calculate the following statistics:

Fraction of virtual cells with a transcription site is calculated by counting how many paired simulations have a total number of weighted nascent RNAs of 2.0 or more. This is done in order to be consistent with the limitations of the experimental data; only nuclear spots with fluorescence greater or equal to two mature mRNA spots were called as transcription sites. When this number of paired simulations is divided by the total of 1000 paired simulations, it is the fraction of virtual cells with a transcription site.

Median number of nascent RNAs per virtual cell is calculated from those paired simulations with a total number of weighted nascent RNAs of 2.0 or more.

### Data and code availability

Experimental analysis code is freely available at https://github.com/bakkerra/smfish_pipeline.

All raw smFISH data after spot and nuclei segmentation is freely available at https://doi.org/10.21985/n2-rfax-bk36. Source data is deposited in the Northwestern University library's data

repository. Each. csv file is for one wing disc analyzed for either nuclei or RNA from a given gene as indicated in each file's name. XYZ centroid positions and fluorescence intensity values are listed.

## Acknowledgements

Fly stocks from Hugo Bellen and the Bloomington Drosophila Stock Center are gratefully appreciated. We thank Jessica Hornick and the Biological Imaging Facility for help with imaging and the Keck Facility at Northwestern for help with probe purification. We are very grateful to Shawn Little and Thomas Gregor for hosting RB at Princeton and their invaluable advice on adapting the smFISH method to imaginal discs. We also thank Arjun Raj and Brian Munsky for key suggestions on experimental and analytical development. Financial support was provided from the NIH (T32CA080621, RB; R35GM118144, RWC), NSF (1764421, MM and RWC), and the Simons Foundation (597491, MM and RWC). MM is a Simons Foundation Investigator.

## Additional information

### Funding

| Funder | Grant reference number | Author |
|---|---|---|
| National Institutes of Health | R35GM118144 | Richard W Carthew |
| National Institutes of Health | T32CA080621 | Rachael Bakker |
| National Science Foundation | 1764421 | Madhav Mani<br>Richard W Carthew |
| Simons Foundation | 597491 | Madhav Mani<br>Richard W Carthew |

The funders had no role in study design, data collection and interpretation, or the decision to submit the work for publication.

### Author contributions

Rachael Bakker, Resources, Data curation, Software, Formal analysis, Visualization, Methodology, Writing - original draft; Madhav Mani, Conceptualization, Formal analysis, Supervision, Funding acquisition, Methodology, Project administration, Writing - review and editing; Richard W Carthew, Conceptualization, Supervision, Funding acquisition, Methodology, Writing - original draft, Project administration

### Author ORCIDs

Richard W Carthew  https://orcid.org/0000-0003-0343-0156

### Decision letter and Author response

Decision letter https://doi.org/10.7554/eLife.56076.sa1
Author response https://doi.org/10.7554/eLife.56076.sa2

## Additional files

### Supplementary files

• Supplementary file 1. Excel file containing the sequences of all oligonucleotide probes used for smFISH experiments in this paper. Each worksheet lists the oligos specific for a gene, as indicated. Sequences are ordered 5' - to - 3'.

• Transparent reporting form

## Data availability

All smFISH data after image segmentation have been deposited in the Public Data Repository at Northwestern University's Library. These data are freely available at https://doi.org/10.21985/n2-rfax-bk36. There are no restrictions.

The following dataset was generated:

| Author(s) | Year | Dataset title | Dataset URL | Database and Identifier |
|---|---|---|---|---|
| Bakker R, Mani M, Carthew RW | 2020 | Data related to Bakker et al 2020 eLife paper | https://doi.org/10.21985/n2-rfax-bk36 | Northwestern University Library Data Repository, 10.21985/n2-rfax-bk36 |

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
