## [Decision Letter]

**Acceptance summary:**

We enjoyed reading this paper and we believe that the wg and dpp data that the authors present here with respect to the regulation of transcription bursts is novel and interesting.

**Decision letter after peer review:**

Thank you for submitting your article "The Wg and Dpp morphogens regulate gene expression by modulating the frequency of transcriptional bursts" for consideration by *eLife*. Your article has been reviewed by two peer reviewers, and the evaluation has been overseen by a Reviewing Editor (Hugo Bellen) and Utpal Banerjee as the Senior Editor. The reviewers have opted to remain anonymous.

The reviewers have discussed the reviews with one another and the Reviewing Editor has drafted this decision to help you prepare a revised submission.

As the editors have judged that your manuscript is of interest, but as described below but some minor additional experiments are required before it is published, we would like to draw your attention to changes in our revision policy that we have made in response to COVID-19 (https://elifesciences.org/articles/57162). First, because many researchers have temporarily lost access to the labs, we will give authors as much time as they need to submit revised manuscripts. We are also offering, if you choose, to post the manuscript to bioRxiv (if it is not already there) along with this decision letter and a formal designation that the manuscript is “in revision at *eLife*”. Please let us know if you would like to pursue this option.

Summary:

The authors have explored mechanism by which two key morphogens regulate the transcription of their downstream targets in *Drosophila* wing imaginal discs. They used Wg and Dpp morphogens to examine kinetic steps during transcriptional initiation that they regulate. They first established single-molecule FISH (smFISH) for some of Wg and Dpp target genes and showed that it has a low false negative and false positive rate. They then performed segmentation and image analysis to assign each smFISH signal to a specific cell, and this allowed them to quantify the number of mRNA molecules per cell. They also showed that the intensity of the smFISH signal can be used to determine whether a given mRNA molecule is nascent or mature and used it to quantify the nascent and mature mRNAs for these genes in each cell. These results suggested that transcription downstream of Wg and Dpp occurs in bursts. To ascertain this notion and to determine which aspect of bursting might be regulated by these morphogens, the authors performed mathematical modeling. Comparison of various simulations with the actual smFISH data allowed the authors to conclude that Wg and Dpp morphogens share a common method for the transcriptional control of their downstream genes, despite the different expression patterns of these targets. The data suggested that these morphogens regulate their targets by modulating the burst frequency not the burst size. Hence, this study connects a specific aspect of transcriptional bursting (i.e. burst frequency) to developmental gene regulatory mechanisms downstream of evolutionarily conserved morphogens.

Essential revisions:

Although linking morphogens to transcriptional burst frequency is exciting, the authors have not made any attempts to find how morphogens regulate burst frequency but not burst size.

The idea championed in this work – that graded expression of targets of morphogen signaling proteins is a product of transcription "burst frequency modulation" – is intriguing and important, and is consistent with recent findings in other contexts. The results and analysis are impressive, and the reviewers advocate for its publication in *eLife*. However a better understanding of how the data was analyzed needs to be provided.

Given that the conclusions the authors draw are likely the predicted outcomes, it is especially important to show a control hybridization experiment that analyzes a gene that does not vary in expression across the disc – one that is regulated by Gal4 perhaps?

In the first paragraph of the results section, "These factors diffuse from their sources forming concentration gradients across the disc, and regulate gene expression in a concentration-dependent manner." The claim that the morphogens diffuse across the disc is inconsistent with recent reports of other mechanisms (filopodia), and as the mechanism of dispersion is irrelevant, this statement is unnecessary.

Figure 1—figure supplement 1A,B: The sens control (B) is uniformly black in the pdf that was provided, but presumably has background spots? Please show the control hybridization of the GFP probe to discs lacking the GFP-sens transgene.

Figure 1—figure supplement 1C: The description for calculating the 3D fluorescent objects is unclear to me. Was the PSF determined, how many sections were imaged, what was the distribution of diffraction-limited spots/of control spots in y? The cut-off that parses signal and background seems arbitrary – is it justified by the distribution detected in the sens and no GFP-sens transgene controls? Please show an intensity graph for these control spots. What are the background spots for intron probes?

Figure 1—figure supplement 2E: I do not understand how this image was generated or what it's meaning might be.

Figure 1—figure supplement 2F: The disc epithelial cells are highly elongated and their morphologies are nothing like the cells depicted in this image. If these theoretical volumes have no actual basis, what is their value? Is there no way (membrane dyes?) to stain cell membranes in the preps? Please at least provide a better description of what "Voronoi tessellation" calculates.

Detecting only one fluorescent spot per nucleus is interesting, especially as it is not what has been observed for the embryo. Embryo nuclei are polarized so that transcription spots are basal for a gene near the telomere. *omb* is near the X telomere – is the *omb* spot polarized with respect to the apicobasal axis? Would it be interesting/informative to compare transcription spots in haploids and diploids? If we predict that the number of nuclei with >1 transcription spot will not be observed, what is the prediction for frequency, for intensity of the single spots?

I am not equipped to evaluate the modeling, so do not appreciate its importance if the number and intensity of transcription spots correlates with the known/observed spatial distributions of RNA and protein in the disc. Again, the methods used to parse the data are not clearly justified and so seem ad hoc.

---

## [Author Response]

Essential revisions:Although linking morphogens to transcriptional burst frequency is exciting, the authors have not made any attempts to find how morphogens regulate burst frequency but not burst size.

Attempts to understand the molecular mechanism of burst frequency modulation by Wg and Dpp will require a major investment in new technologies that are currently available only in the early fly embryo (Levine, Gregor, Garcia groups). There, bursts are visualized in live embryos by MS2-tagged nascent RNA. Visualizing transcription factor binding is still in its infancy, notably Darzacq achieving single molecule imaging in living *Drosophila* embryos using lattice light-sheet microscopy – or using the ParS/B system developed by Gregor. Comparable genetic tools will need to be built and tested for the wing disc, conditions for culturing wing discs ex vivo that retain normal development will need to be worked out, and an analytical pipeline will need to be built before experimentation in those directions can begin. Thus, such attempts are well beyond the scope of this present work.

The idea championed in this work – that graded expression of targets of morphogen signaling proteins is a product of transcription "burst frequency modulation" – is intriguing and important, and is consistent with recent findings in other contexts. The results and analysis are impressive, and the reviewers advocate for its publication in eLife. However a better understanding of how the data was analyzed needs to be provided.

We thank the reviewers for pointing out our lack of clarity in describing how the data was analyzed. We have addressed this issue and improved the manuscript’s clarity. In particular we have moved some of the description of how we analyzed data from the Materials and methods section to the main text and figure legends to better aid the reader about key issues. We retain in the Materials and methods, a full and comprehensive description of the data analysis.

Given that the conclusions the authors draw are likely the predicted outcomes, it is especially important to show a control hybridization experiment that analyzes a gene that does not vary in expression across the disc – one that is regulated by Gal4 perhaps?

We have performed smFISH on the *scalloped* (*sd*) gene, whose expression is uniform across the wing pouch (Campbell et al., 1992; Williams et al., 1993). This data is now shown in Figure 2—figure supplement 1, and discussed in the text. The data clearly shows that the average *sd* mRNA/cell is constant across the disc, the likelihood of detecting a transcription site is also constant across the disc, and the nascent RNA number is constant across the disc. It demonstrates that the results with the other genes are not artifacts of variable smFISH or errors in the analytical pipeline.

In the first paragraph of the results section, "These factors diffuse from their sources forming concentration gradients across the disc, and regulate gene expression in a concentration- dependent manner." The claim that the morphogens diffuse across the disc is inconsistent with recent reports of other mechanisms (filopodia), and as the mechanism of dispersion is irrelevant, this statement is unnecessary.

We have modified the statement and deleted claims of diffusion.

Figure 1—figure supplement 1A,B: The sens control (B) is uniformly black in the pdf that was provided, but presumably has background spots? Please show the control hybridization of the GFP probe to discs lacking the GFP-sens transgene.

Panel B did show control hybridization of GFP probe to a disc lacking the sens-GFP gene. We have made that more explicit. Shown in Figure 1—figure supplement 1A,B were raw images taken using the same conditions in which diffraction-limited spots were readily apparent by eye for GFP mRNAs in 1A. We have artificially increased the intensity of all pixels in the images four-fold, so that the reviewer and readers can see the background. They are now added along with the original images to Figure 1—figure supplement 1A,B.

Figure 1—figure supplement 1C: The description for calculating the 3D fluorescent objects is unclear to me. Was the PSF determined, how many sections were imaged, what was the distribution of diffraction-limited spots/of control spots in y? The cut-off that parses signal and background seems arbitrary – is it justified by the distribution detected in the sens and no GFP-sens transgene controls? Please show an intensity graph for these control spots. What are the background spots for intron probes?

We apologize for the lack of details in the figure legend and main text. We have rewritten the section text and figure legend to make it clearer. We had already described in detail the method to calculate the objects in the Materials and methods but this has also been rewritten to improve clarity. The cutoff is chosen based on a characteristic plateau from a parameter sweep of cutoff values performed on each image stack (see Figure 1C). This method of cutoff choice is based on two QC criteria: (1) it IDs spots with <5% error compared to a manually curated ground truth; (2) it identified 200-fold fewer spots in GFP null control discs when compared to GFP-sens discs. There are so few background control spots (~40) that an intensity graph is not helpful. Background for the intron probe is comparable to the exonic probes.

Figure 1—figure supplement 2E: I do not understand how this image was generated or what it's meaning might be.

This is 5 contiguous z-sections of segmented nuclei colored and viewed laterally using the ImageJ 3D viewer plugin. It is not intended for any analysis, only to illustrate the three-dimensional “stack of pancakes” nature of the nuclear objects. We have provided more clarity and information in the figure legend.

Figure 1—figure supplement 2F: The disc epithelial cells are highly elongated and their morphologies are nothing like the cells depicted in this image. If these theoretical volumes have no actual basis, what is their value? Is there no way (membrane dyes?) to stain cell membranes in the preps? Please at least provide a better description of what "Voronoi tessellation" calculates.

We had previously tried to use membrane dyes to visualize cell boundaries but these failed to give adequate results. Exhaustive attempts using many experimental means of visualizing membranes were failures.

A Voronoi tessellation takes a grid of some finite number of points in either 2D or 3D. In our case, each point is the centroid of an imaged 3D nucleus. For each centroid, there is a corresponding region consisting of all other points of the plane that are closer to that centroid than to any other centroid. These regions are called Voronoi cells. The boundaries between Voronoi cells represent points that are equidistant between two centroids. These are taken to represent virtual cell boundaries. It is important to be clear that the Voronoi cells do not accurately describe the pseudostratified epithelial nature of wing disc cells. However, note that 3D segmentation of pseudostratified epithelial cells is still something no one working in *any* system has achieved.

Tessellation is just a way to democratically assign mature transcripts to cells based on vicinity to the nucleus. Clearly we get the transcript assignment incorrect at a local level. However, being the most democratic approach, the trends of mRNAs/cell assigned to 100s of cell across the wing disc are trustworthy. The same logic has been used by Gregor and others assigning mRNA transcripts to early embryonic nuclei when cell boundaries are unseen (e.g., Little et al., 2013). We have now added a clearer description of the tessellation and its meaning in the text and figure legend.

Detecting only one fluorescent spot per nucleus is interesting, especially as it is not what has been observed for the embryo. Embryo nuclei are polarized so that transcription spots are basal for a gene near the telomere. omb is near the X telomere – is the omb spot polarized with respect to the apicobasal axis?

Calculating the z-plane difference between each *omb* transcription site centroid and its corresponding nuclear centroid shows a distribution symmetrically clustered around a mean of zero. This suggests that there is no apical-basal polarity in location of *omb* transcription sites relative to the nucleus can be detected in this data. We ran this analysis for all of the genes, and the distributions look identical to *omb*.

**Author response image 1. sa2fig1:** 

Would it be interesting/informative to compare transcription spots in haploids and diploids? If we predict that the number of nuclei with >1 transcription spot will not be observed, what is the prediction for frequency, for intensity of the single spots?

Although there are no haploid cells in imaginal disc tissue, we did compare transcription sites with one copy versus two copies of the *sfGFP-sens* transgene. We used probes specific for sfGFP RNA.

Animals with 1 copy also bear one fully functional copy of the endogenous *sens* gene. When only one copy of the transgene is present, the likelihood of detected transcription sites is decreased across the disc (roughly by half) (see Author response image 2). The median number of nascent RNAs detected per transcription site is statistically unchanged (see Author response image 2). Bootstrapped 95% confidence intervals overlap, and we did a permutation test for significance which gave a p-value of 0.3. These observations are consistent with an interpretation that in cells with two *sfGFP-sens* copies, the observed nascent RNAs represent transcription from a single allele at any given time. Halving the number of alleles decreases the likelihood of observing a transcription burst, but does not strongly affect the number of RNAs observed per burst. Given that the likelihood of observing a site for a single *sfGFP-sens* allele is maximally 10%, the likelihood of observing two sites at the same time would be at most 1%, assuming independent firing.

However, for genes probed far upstream of the transcription termination site, it is very likely that we are seeing transcription from both alleles at once, given that a detectable nascent RNA would stay at the transcription site for a long time (~50 minutes). Even for these very bright transcription spots, only one transcription site per nucleus is observed (clearly visible in Figure 3 B and C). This observation, and the literature suggesting that allelic pairing occurs extensively in *Drosophila* tissue, argues that we still need to assume that transcription from a second allele would not appear as a second transcription site.

We have added this commentary to the revised manuscript text.

Permutation test methods: Data from two experimental groups was pooled and repeatedly randomly shuffled into two new groups. The median of each random group was calculated and the difference between these two groups recorded. After 10,000 reshufflings, 30% of the randomly assigned groups had a difference in median which was equal to or greater than that observed in the actual data. This indicates that the two groups are not significantly different.

I am not equipped to evaluate the modeling, so do not appreciate its importance if the number and intensity of transcription spots correlates with the known/observed spatial distributions of RNA and protein in the disc. Again, the methods used to parse the data are not clearly justified and so seem ad hoc.

As previously stated, we have now made the methods of data analysis more clear and have taken pains to explain how they relate to modeling. Summarizing, we are able to make 3 measurements at single-cell resolution from our smFISH measurements: (1) the number of mature transcripts in the cytoplasm of cells, (2) the presence/absence of nascent sites of transcription in cell nuclei, and (3) the intensity of nascent RNAs. These 3 empirical observables are downstream of the kinetics of the promoter, which is what we wish to have insight into. The model forms a bridge from promoter kinetics to these three observables. The simulations of the model make manifest that the mapping from promoter kinetics to our observables is sufficiently complex, making intuitive rationalizing of the trends challenging.